# Structural insights into recognition of chemokine receptors by *Staphylococcus aureus* leukotoxins

Paul Lambey[1†], Omolade Otun[1†], Xiaojing Cong[1], François Hoh[2], Luc Brunel[3], Pascal Verdié[3], Claire M Grison[1], Fanny Peysson[1], Sylvain Jeannot[1], Thierry Durroux[1], Cherine Bechara[1,4], Sébastien Granier[1*], Cédric Leyrat[1*]

[1]Institut de Génomique Fonctionnelle, Université de Montpellier, CNRS, INSERM, Montpellier, France; [2]Centre de Biochimie Structurale, CNRS UMR 5048-INSERM 1054- University of Montpellier, Montpellier, France; [3]Institut des Biomolécules Max Mousseron (IBMM), University of Montpellier, Montpellier, France; [4]Institut Universitaire de France, Paris, France

**Abstract** *Staphylococcus aureus* (SA) leukocidin ED (LukED) belongs to a family of bicomponent pore forming toxins that play important roles in SA immune evasion and nutrient acquisition. LukED targets specific G protein-coupled chemokine receptors to lyse human erythrocytes (red blood cells) and leukocytes (white blood cells). The first recognition step of receptors is critical for specific cell targeting and lysis. The structural and molecular bases for this mechanism are not well understood but could constitute essential information to guide antibiotic development. Here, we characterized the interaction of LukE with chemokine receptors ACKR1, CCR2, and CCR5 using a combination of structural, pharmacological, and computational approaches. First, crystal structures of LukE in complex with a small molecule mimicking sulfotyrosine side chain (p-cresyl sulfate) and with peptides containing sulfotyrosines issued from receptor sequences revealed the location of receptor sulfo-tyrosine binding sites in the toxins. Then, by combining previous and novel experimental data with protein docking, classical and accelerated weight histogram (AWH) molecular dynamics we propose models of the ACKR1-LukE and CCR5-LukE complexes. This work provides novel insights into chemokine receptor recognition by leukotoxins and suggests that the conserved sulfotyrosine binding pocket could be a target of choice for future drug development.

## Editor's evaluation

In this report, the authors employed a wide array of biophysical tools and techniques to study the interaction of LukE toxin with chemokine receptors. Although follow-up studies will be required to substantiate all of the conclusions drawn by the authors, the paper will be of general interest as it provides insights into the molecular recognition mechanism of an important toxin interacting with cellular receptors.

## Introduction

*Staphylococcus aureus* (SA) is a major opportunistic human pathogen that causes a wide range of clinical manifestations (***Tong et al., 2015***), and poses growing health concern due to the emergence of multidrug resistant strains. Pathogenic SA strains produce a number of wall-associated or secreted virulence factors that promote growth, nutrient acquisition and evasion of the host immune system (***DeLeo et al., 2009***; ***Oliveira et al., 2018***). Among this arsenal of virulence factors, the bicomponent

*For correspondence:
sebastien.granier@igf.cnrs.fr (SG);
cedric.leyrat@igf.cnrs.fr (CL)

†These authors contributed equally to this work

Competing interest: The authors declare that no competing interests exist.

leukotoxins are β-barrel pore forming toxins that play an important role in SA pathogenesis by targeting host leukocytes and erythrocytes (*Spaan et al., 2017*), and appear to be promising targets for drug development (*Kong et al., 2016*). Human clinical SA isolates can produce up to five different leukotoxins: γ-hemolysin AB and CB (HlgAB and HlgCB), Leukocidin ED (LukED), Panton-Valentine Leukocidin (PVL or LukSF-PV), and Leukocidin AB (LukAB, also known as LukGH). Each leukotoxin is made up of two subunits of approximately 30 kDa: a host cell targeting S-component (HlgA, HlgC, LukE, lukS-PV, and LukA/G), and a polymerization F- component (HlgB, LukD, lukF-PV, and LukB/H), which can bind the plasma membrane through highly conserved phosphocholine binding sites (*Olson et al., 1999*; *Liu et al., 2020*). S- and F-components typically display relatively high conservation within each group ( ~ 70% sequence identity), but sequence identity drops below 30% between groups (*Spaan et al., 2017*).

With the exception of LukGH, these toxins are secreted as monomers that bind target cells and subsequently octamerize into a lytic pore on the plasma membrane, composed of alternating S- and F- subunits (*Olson et al., 1999*; *Guillet et al., 2004*; *Yamashita et al., 2011*; *Yamashita et al., 2014*; *Nocadello et al., 2016*; *Trstenjak et al., 2020*). Each toxin is composed of two domains: a central β sandwich called the CAP domain and involved in inter-protomer interactions, and a RIM domain which contains several divergent loops driving host cell specificity. The STEM region is a subdomain of the CAP, which changes its conformation upon pore formation to associate with neighboring protomers and form a β barrel that inserts into the cell membrane (*Yamashita et al., 2011*; *Yamashita et al., 2014*).

Over the last decade, membrane protein receptors have been identified for most of the bicomponent leukotoxins, providing key insights into their role in SA pathogenesis and the mechanisms driving their cellular tropism and species specificity (*Spaan et al., 2017*). Specifically, HlgA and LukE were shown to share atypical chemokine receptor 1 (ACKR1) as their main receptor on erythrocytes (*Spaan et al., 2015*), while recruiting CCR5 (LukE), CCR2 (HlgA), CXCR1, and CXCR2 (both HlgA and LukE) to differentially target specific leukocyte populations (*Alonzo et al., 2013*; *Reyes-Robles et al., 2013*; *Spaan et al., 2014*). Conversely, HlgC and LukS-PV were found to bind monocytes and neutrophils through C5aR1 and C5aR2 (*Spaan et al., 2013*; *Spaan et al., 2014*). All of the identified receptors belong to the chemokine or complement receptor families of class A G-protein-coupled receptors (GPCRs), except for the divergent lukGH toxin which binds CD11b (aka complement receptor 3) expressed on phagocytic cells (*DuMont et al., 2013*). Recently, F-component-specific receptors have been identified as well: human CD45 was shown to act as a receptor for lukF-PV (*Tromp et al., 2018*), while HlgB was found to interact with ACKR1.

Several of these toxin-GPCR pairs have been characterized at the molecular level by swapping mutagenesis of divergent regions/loops (DRs) of the RIM domain, in particular for LukE and HlgA that can target multiple chemokine receptors (*Reyes-Robles et al., 2013*; *Laventie et al., 2014*; *Tam et al., 2016*; *Peng et al., 2018*; *Vasquez et al., 2020*). These studies highlighted the critical role of several DRs located in the rim domain in determining receptor specificity. From the receptor's side, residues of CCR5 were identified in eCL2 and in the upper part of TM7 that lead to severe loss of LukED activity (*Tam et al., 2016*). Several residues of ACKR1 were also identified by mutagenesis, the most critical being the sulphated tyrosine sTyr41 for which the alanine substitution nearly abolished LukED and HlgAB activity on erythrocytes (*Spaan et al., 2015*). Interestingly, all of the GPCRs that bind leukotoxins possess between 1 and 4 potentially sulfated tyrosine residues in their N-terminal regions, and the interaction between LukS-PV and the C5aR1 N-terminal region was shown to require tyrosine sulfation (*Spaan et al., 2013*). In addition, post-translational modification pathways involved in the sulfation of the leukotoxin receptors were recently found to impact on HlgAB, HlgCB, LukED, and PVL induced cytotoxicity (*Tromp et al., 2020*). Despite these advances, a structural understanding of toxin–chemokine receptors interactions is still lacking due to the unavailability of high resolution structures of the complexes.

In the present study, we provide a detailed characterization of the structure and dynamics of monomeric LukE using integrated structural biology methods. We show that recombinant LukE is able to displace bound chemokines from ACKR1, CCR5, and CCR2 on the cell surface using a time-resolved fluorescence energy transfer (TR-FRET) based approach. We next solved the crystal structures of LukE in complex with a small molecule mimicking a sulfotyrosine side chain (p-cresyl sulfate) and with sulfopeptides derived from the N-terminal region of ACKR1 and CCR2, revealing two conserved

sulfotyrosine binding sites. Finally, we use computational docking and molecular dynamics (MD) simulations to propose models of the ACKR1-LukE and CCR5-LukE complexes that are supported by previous pharmacological experiments as well as novel mutagenesis data described here.

## Results

### Solution and crystallographic structures of LukE

The crystal structure of monomeric LukE has previously been reported at 3.2 Å resolution (PDB code 3ROH) (*Nocadello et al., 2016*). While attempting to co-crystallize LukE in complex with peptides encompassing the sulfated ACKR1 Tyr41 residue, we serendipitously obtained two additional apo crystal forms of LukE at 1.5 Å (apo1) and 1.9 Å (apo2), allowing us to build a higher quality model (see *Table 1* and Materials and methods). The apo2 structure, like the original 3ROH structure, belongs to space group I4, while the apo1 structure crystallized in $P2_12_12_1$ and features an entirely different crystal packing (*Figure 1A* and *Figure 1—figure supplement 1*). All three structures contain one copy of the protein in the asymmetric unit and display very similar conformations, with Cα RMSDs of 1.18 Å (apo1) and 0.81 Å (apo2) relative to the 3ROH structure (*Figure 1A*). One notable difference is the presence of residues 12–23 belonging to the N-terminal signal sequence in apo2, which are involved in crystal packing interactions, possibly accounting for the higher resolution compared to the 3ROH structure (*Figure 1A* and *Figure 1—figure supplement 1*). A comparison of the B-factors of the three structures is shown in *Figure 1—figure supplement 1D-F*, highlighting the influence of crystal packing on protein flexibility within the crystals.

In order to obtain structural information about LukE directly in solution, we used SAXS. The experimental SAXS profiles are shown in *Figure 1B*. The samples were free from aggregation as evidenced by the linearity of the Guinier region, however, the measured radius of gyration (Rg) increased from 2.41 ± 0.02 nm at 1 mg/ml to 2.68 ± 0.08 at 4 mg/ml of protein, suggesting a slight tendency for interparticle attraction. This was consistent with a native mass spectrum of LukE showing a small proportion of dimeric forms in equilibrium with LukE monomers (*Figure 1—figure supplement 2*). In order to extract structural information from the SAXS data, we turned to the ensemble optimization method (EOM) using ensembles of models derived from atomistic MD simulations (*Figure 1C, D*, and *Figure 1—figure supplement 1G-I*). The MD generated models of LukE were fitted to a merged SAXS curve to remove the interparticle attraction effects (*Figure 1B*), yielding a $\chi$ value of 0.79. Rg values of the selected models were in a narrow range between 2.40 and 2.55 nm (*Figure 1D*), consistent with the values of 2.32 and 2.43 nm calculated from the apo1 and apo2 structures. Conformations of the models from the optimized ensemble show a very rigid core, with highly flexible terminal regions corresponding to the signal peptide and polyhistidine tag. Localized flexibility is also present in the divergent loops of the RIM domain, and in the STEM region loop (residues 151–159) (*Figure 1B* and *Figure 1—figure supplement 1*). The three β strands of the STEM remain, however, relatively rigid, suggesting that specific conformational triggers are required to explore the extended conformations observed in leukotoxin pore structures.

### LukE competes with CCL5 binding onto ACKR1 and CCR5, but also with CCL2 binding onto CCR2

We next sought to analyze the binding of LukE to its human chemokine receptor targets. A wealth of experimental information is available indicating that the expression of CCR5, ACKR1, CXCR1, and CXCR2 render human cells susceptible to LukED killing (*Alonzo et al., 2013*; *Reyes-Robles et al., 2013*; *Spaan et al., 2015*; *Tam et al., 2016*; *Vasquez et al., 2020*). Direct interaction between LukE and CCR5 or ACKR1 has also previously been characterized by surface plasmon resonance (SPR) in purified systems, indicating Kd values around 40 and 60 nM, respectively (*Alonzo et al., 2013*; *Vasquez et al., 2020*). In order to complement the available experimental data and to verify that our recombinantly produced LukE is biologically active, we performed competitive binding assays in HEK293T cells using Homogenous TR-FRET technology (*Zwier et al., 2010*; *Figure 2*). For these experiments, we selected ACKR1 and CCR5, which are major leukotoxin receptors in erythrocytes and leukocytes, but also CCR2 which is normally an in vivo target of HlgA rather than LukE (*Spaan et al., 2014*). SNAP-tag-fused CCR5, ACKR1, and CCR2 receptors transiently expressed in HEK293 cells were covalently labeled with Lumi4-terbium as donor. Cells were then incubated in the presence

**Table 1.** X-ray data collection and refinement statistics (molecular replacement).

| | Apo1 | Apo2 | p-cresol sulfate soak | AcDsYDsYG-NH2 soak | AcDSFPDGDsY GANLE-NH2 soak |
|---|---|---|---|---|---|
| **Data collection** | | | | | |
| Space group | P2$_1$2$_1$2$_1$ | I4 | P2$_1$2$_1$2$_1$ | P2$_1$2$_1$2$_1$ | P2$_1$2$_1$2$_1$ |
| **Cell dimensions** | | | | | |
| a, b, c (Å) | 62.58, 72.20, 78.81 | 135.79, 135.79, 63.53 | 62.58, 73.49, 79.34 | 63.30, 72.41, 79.00 | 62.97, 71.18, 79.23 |
| a, b, g (°) | 90.00, 90.00, 90.00 | 90.00, 90.00, 90.00 | 90.00, 90.00, 90.00 | 90.00, 90.00, 90.00 | 90.00, 90.00, 90.00 |
| Resolution (Å) | 49.01–1.46 (1.48–1.46) | 48.01–1.90 (1.94–1.90) | 47.65–1.60 (1.63–1.60) | 49.40–1.40 (1.42–1.40) | 47.16–1.55 (1.58–1.55) |
| $R_{merge}$ | 0.079 (2.601)[NA] | 0.100 (1.510)[NA] | 0.040 (0.710) | 0.038 (1.129)[NA] | 0.032 (0.811) |
| $CC_{1/2}$ | 1.000 (0.508) | 0.984 (0.748) | 0.996 (0.624) | 0.999 (0.483) | 0.998 (0.622) |
| $I / \sigma I$ | 17.2 (1.0) | 19.2 (1.9) | 15.5 (1.7) | 15.8 (1.2) | 17.5 (1.5) |
| Completeness (%) | 99.9 (98.5) | 100 (100) | 99.3 (99.2) | 99.2 (97.3) | 99.2 (99.8) |
| Redundancy | 13.1 (12.0) | 13.7 (13.8) | 3.4 (3.1) | 4.1 (3.6) | 3.2 (3.3) |
| **Refinement** | | | | | |
| Resolution (Å) | 49.1–1.46 | 48.1–1.90 | 47.7–1.60 | 40.9–1.40 | 47.2–1.55 |
| No. reflections | 62,775 | 45,734 | 48,967 | 72,136 | 52,377 |
| $R_{work}$ / $R_{free}$ | 17.81/19.41 | 18.63/20.30 | 18.12/20.46 | 18.44/19.81 | 18.12/20.37 |
| **No. atoms** | | | | | |
| Protein | 2,370 | 2,472 | 2,351 | 2,359 | 2,276 |
| Ligand/peptide | 0 | 0 | 57 | 71 | 140 |
| Water | 328 | 323 | 260 | 293 | 215 |
| **B-factors** | | | | | |
| Protein | 28.3 | 37.5 | 32.8 | 31.7 | 36.9 |
| Ligand/peptide | NA | NA | 29.6 | 31.8 | 38.3 |
| Water | 44.3 | 53.3 | 48.4 | 47.8 | 51.1 |
| **R.m.s. deviations** | | | | | |
| Bond lengths (Å) | 0.009 | 0.009 | 0.010 | 0.010 | 0.010 |
| Bond angles (°) | 1.108 | 1.092 | 1.102 | 1.226 | 1.163 |
| <Ligand occupancy> | NA | NA | 0.67 | 0.59 | 0.51 |

Values in parentheses are for highest-resolution shell. NA–not applicable, $R_{merge}$ value over 1 is statistically meaningless.

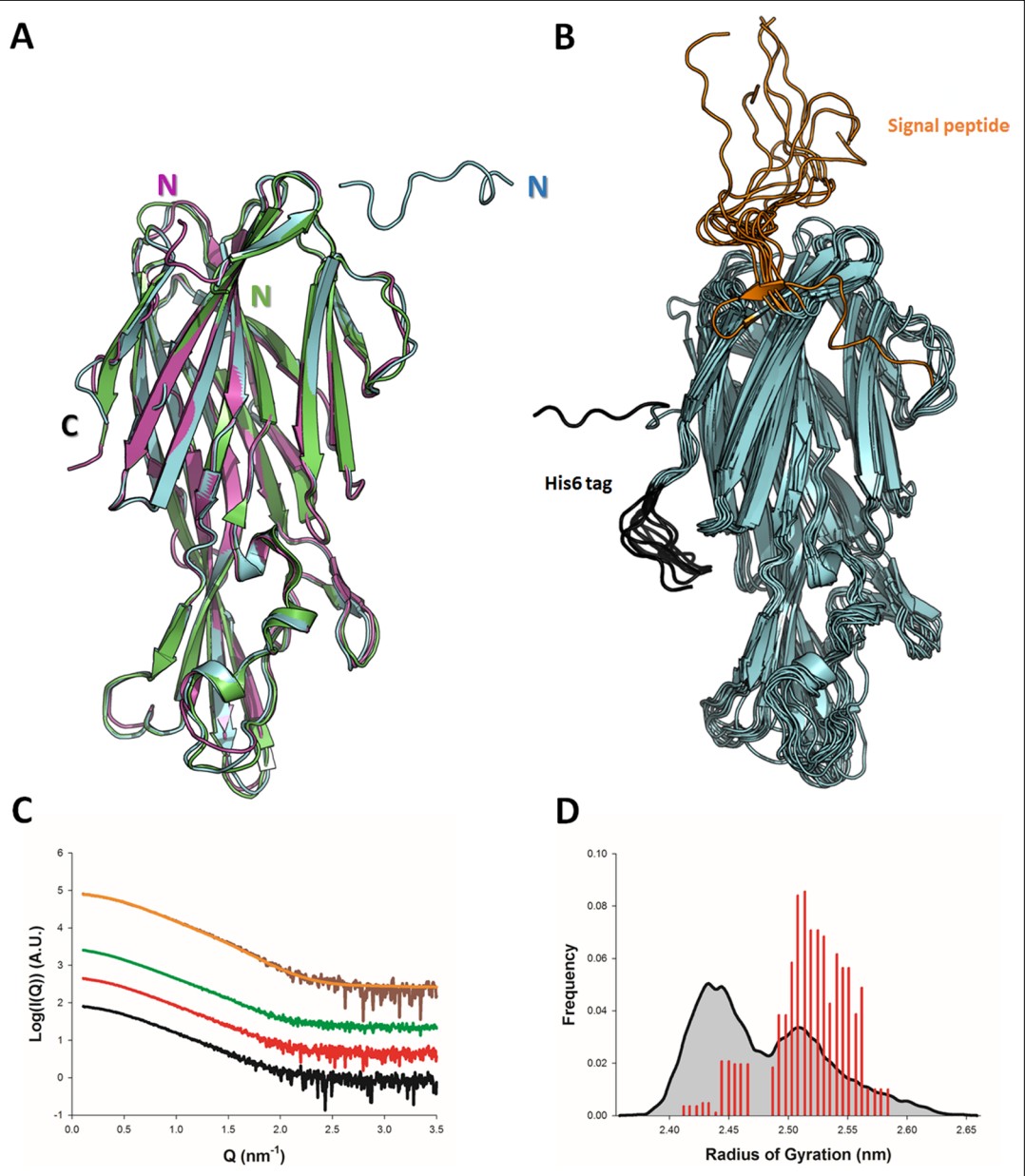

**Figure 1.** Solution and crystallographic structures of leukotoxin LukE. (**A**) The crystal structures of LukE in P2$_1$2$_1$2$_1$ (Apo 1, green) and I4 (Apo 2, cyan) space groups are shown in cartoon representation and overlaid onto the previously published crystal structure of LukE (magenta), also in I4 space group (PDB code 3ROH). (**B**) Optimized ensemble of 10 models corresponding to the fitted SAXS profile in C. The N-terminal extension belonging to the signal peptide and the C-terminal polyhistidine tag are shown in orange and black, respectively. The CAP, RIM, and STEM domains are colored in cyan, deep blue and light blue, respectively. Other regions of interest are also indicated such as the D-region, and divergent loops 1–4. (**C**) Small angle x-ray scattering (SAXS) profiles of LukE measured at 1, 2, and 4 mg/ml are shown as black, red and green lines, respectively. The merged SAXS curve is shown as a brown line with the fitting curve obtained using the ensemble optimization method (EOM) in orange. (**D**) Radius of gyration distributions for the initial pool ensemble (gray area and black line) and for the optimized ensembles (red bars) obtained using the merged SAXS curve.

The online version of this article includes the following figure supplement(s) for figure 1:

**Figure supplement 1.** Crystallographic packing differences between LukE structures, LukE flexibility in crystal structures and molecular dynamics simulations.

**Figure supplement 2.** Native MS spectrum of LukE.

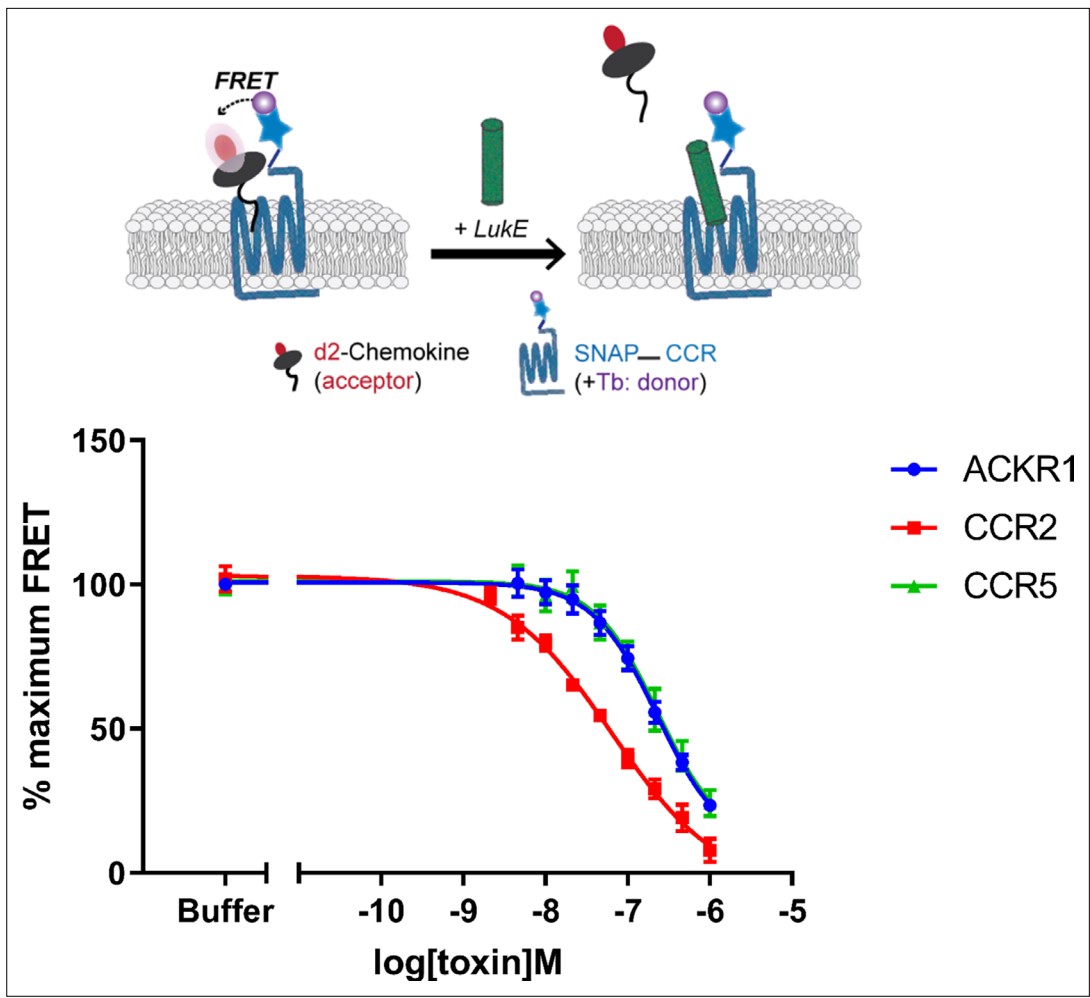

**Figure 2.** Binding of LukE in live cells as determined by competition time-resolved fluorescence energy transfer (TR-FRET). The upper panel shows a schematic of the competitive TR-FRET assay. Addition of toxins disrupts energy transfer between a SNAP-tagged receptor labeled with a Terbium donor and a d2-chemokine acceptor (*Zwier et al., 2010*). The bottom panel shows the competition dose-response curves at receptors CCR5, ACKR1, and CCR2. 5 nM tracer ligands, CCL5-d2 for CCR5 and ACKR1 and CCL2-d2 for CCR2 were used to determine TR-FRET at their respective receptors in the presence of LukE. IC50 values are quoted in-text. Data shown is mean ± SEM of three independent experiments performed in triplicate.

The online version of this article includes the following source data for figure 2:

**Source data 1.** Raw data of the LukE competition TR-FRET assay.

of 5 nM tracer chemokines d2-CCL5 (CCR5 and ACKR1) or d2-CCL2 (CCR2). Addition of increasing concentrations of LukE led to a decrease in the TR-FRET ratio for all receptors indicating the ability of LukE to bind competitively with IC50s of 231.7 nM ±0.19, 230.4 nM ±0.14 and 64.2 nM ±0.14 at CCR5, ACKR1, and CCR2, respectively. These results indicate at least partial overlap between the LukE and CCL2/5 binding sites on the receptors extracellular surfaces and are consistent with previous observations regarding ACKR1- and CCR5- LukE interactions (*Alonzo et al., 2013*; *Spaan et al., 2015*). Interestingly, LukE inhibited CCR2–CCL2 interaction with a better IC50 than observed for CCR5–CCL5 and ACKR1–CCL5 pairs, suggesting a strong interaction between CCR2 and LukE.

## Crystal structure of LukE in complex with p-cresyl sulfate reveals three potential sulfotyrosine binding sites

In order to obtain structural insights into the interaction of LukE with its receptors, we carried out crystal soaking experiments using apo1 LukE crystals. Because recognition of sulfated tyrosines

appears to be a common trend among leukotoxins, we selected 3 synthetic molecules for soaking, in an attempt to identify potential sulfotyrosine binding sites: (1) para-cresyl sulfate (pCS) is a metabolite of tyrosine and a uremic toxin (*Gryp et al., 2017*). Its chemical structure is identical to that of a sulfotyrosine side chain (Cβ replaced by a methyl group); (2) the [34]DSFPDGDsYGANLE[46] peptide of ACKR1, with a functionally critical sulfated tyrosine in position 41; and (3) the [25]DsYDsYG[29] peptide of CCR2 bearing 2 sulfated tyrosines in position 26 and 28. Although CCR2 is normally an in vivo target of HlgA rather than LukE, Tam and coworkers have shown that cells expressing WT CCR5 or a CCR5 chimera harbouring the CCR2 N-terminal region display the same susceptibility to LukED intoxication (*Tam et al., 2016*). This interchangeability of the N-terminal region is perhaps not so surprising given that both sequences contain multiple tyrosine residues that can be efficiently sulfated (*Farzan et al., 1999*; *Preobrazhensky et al., 2000*; *Tan et al., 2013*). Crystals soaked with all three compounds yielded good diffraction data (*Table 1*) that revealed strong additional electron density corresponding to the ligands (*Figure 3—figure supplement 1* and *Figure 3—figure supplement 2*).

The structure of pCS-bound LukE is shown in *Figure 3*. Three pCS molecules interact with LukE at distinct sites, which we named site 1, 2, and 3. Site 1 forms a surface pocket with Arg263 sidechain sitting at the bottom and engaging in a salt bridge with the pCS sulfate group (*Figure 3C*). Arg85 and Arg290 guanidinium groups additionally flank the sulfate oxygens, and the pCS aromatic ring packs against the hydrophobic sidechains of Ile103, Leu265, and Phe287. The upper part of the pocket is delimited by Lys52 and Trp53, which shield the pCS interacting arginines from the solvent. These residues correspond to Arg24 and Leu25 in HlgA and are part of the D-region that plays an important role in hemolytic activity (*Nariya and Kamio, 1997*; *Peng et al., 2018*).

Site 2 is a relatively flat area located approximately 0.8 nm below site 1 (*Figure 3D*). The pCS sulfate is stabilized by polar contacts with Tyr269 and Arg101, and Arg101 additionally stacks its guanidinium group against the pCS phenyl ring, engaging in cation–π interaction. Ile103 and Phe287 from site 1 are also involved in hydrophobic contacts with the pCS phenyl ring. Ser252 from a crystallographically related molecule further stabilizes pCS binding through polar contacts with the sulfate group.

Site 3 corresponds to a small groove in between Lys 185, Asn109, Asn291, and Ser260 sidechains (*Figure 3B*). The pCS molecule is mainly stabilized by polar interactions of the sulfate group with with Lys 185, Asn109, and Ser260, as well as hydrophobic contact between the phenyl ring and the aliphatic part of Lys185 sidechain.

## Crystal structures of LukE in complex with the sulfated peptides [34]DSFPDGDsYGANLE[46] of ACKR1 and [25]DsYDsYG[29] of CCR2 show sulfotyrosine binding at sites 1 and 2

The structure of LukE in complex with the ACKR1 peptide [34]DSFPDGDsYGANLE[46] is shown in *Figure 4A*. Unexpectedly, we found two bound copies of the peptide, which may be due to the potentially very high peptide concentration used during crystal soaking experiments (see Materials and methods). Residues 34–43, and residues 39–46 are visible for the 1st and 2nd peptide, respectively. Although the signal for each sulfotyrosine is fairly strong, the electron density of the other peptide residues is weaker, particularly for the 1st peptide (see Polder map in Figure S4B). This is apparent in the refined occupancies which are ~0.7 for the sulfotyrosines, but only 0.3–0.6 for the other peptide residues. The first peptide interacts with site 1 through its sTyr41 residue in a manner similar to the pCS molecule. The preceding residues [36]FPDGD[40] close the back of the binding pocket, decreasing the solvent exposed surface area of sTyr41, and provide additional stability through multiple intermolecular interactions (*Figure 4C*). Phe36 and Pro37 pack hydrophobically against Lys51 and Lys52 of LukE. Asp38 forms a salt bridge with LukE Arg290, and Gly39 carbonyl oxygen is hydrogen bonded to Arg101 from site 2. The 2nd peptide sTyr41 residue interacts with site 2, in a way that is also similar to pCS, although Arg101 sidechain adopts a different conformation and an additional salt bridge is formed between the sulfate and Lys283 (*Figure 4E*). Other stabilizing interactions include several backbone-backbone hydrogen bonds between Asp40-Lys92, Gly42-Asp90, and Leu45-Phe88, and hydrophobic interactions between Leu45 and Phe88, Ala219 and Arg220 of LukE. Although the conformation of both peptides is stabilized by multiple interactions with LukE, it should be noted that the 2nd peptide chain runs with its C-terminus going away from the putative location of the membrane. In addition, the occupancy is relatively low (except for the sulfotyrosines) and the conformation of the peptides is likely influenced by crystal packing and the associated steric constraints.

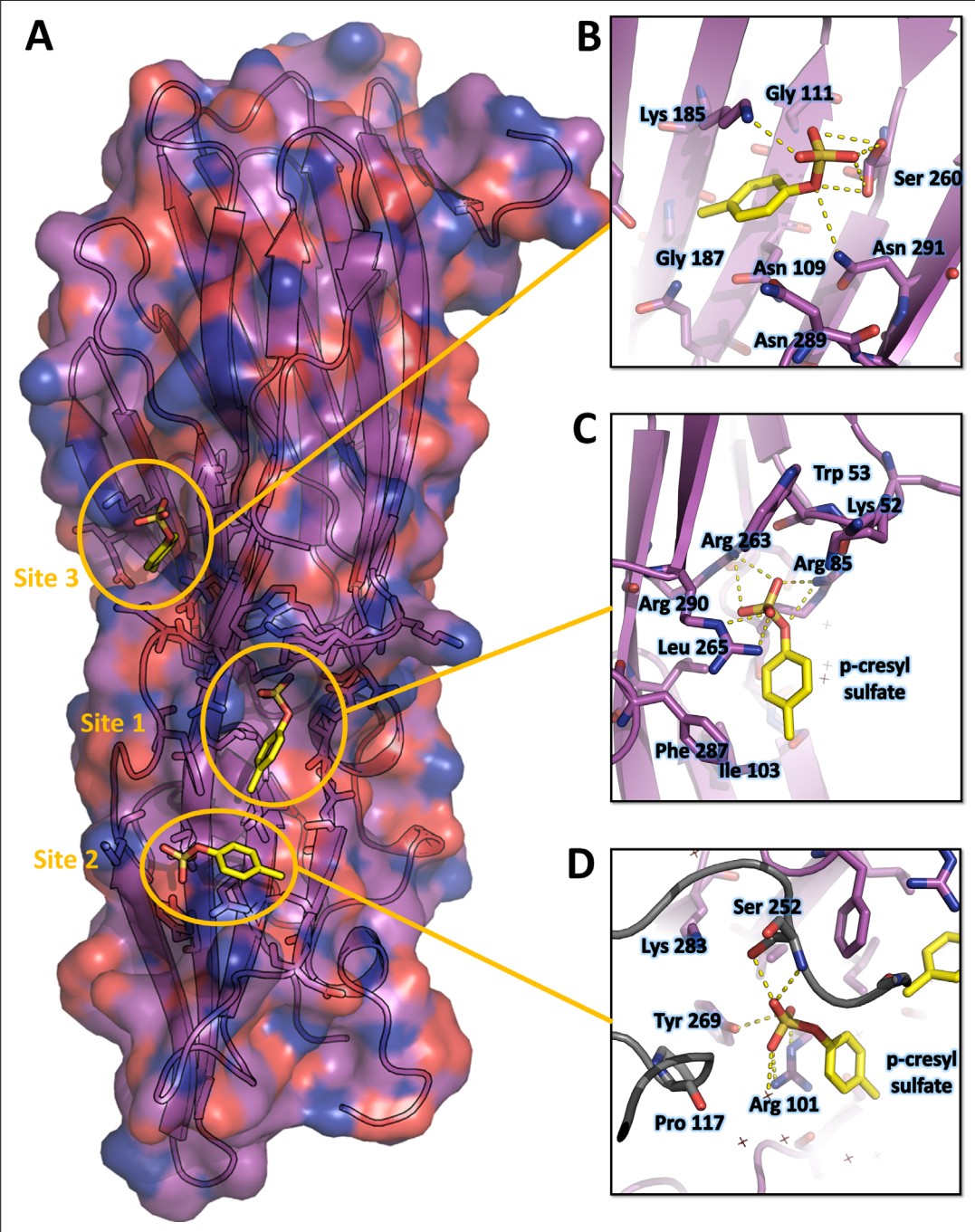

**Figure 3.** Crystal structure of LukE in complex with p-cresyl sulfate. (**A**) Overview of the asymmetric unit of the crystal. Protein is shown in cartoon representation and colored magenta. The p-cresol sulfate molecules are shown in yellow sticks with surrounding protein residue sidechains in magenta lines (**B–D**). Close ups of the sulfotyrosine binding sites. The color code is the same as in (**A**) with p-cresol sulfate molecules and protein sidechains shown in sticks. Polar contacts between ligand and protein are shown as yellow dashed lines. In (**D**), protein residues from an interacting symmetry related molecule are shown in gray cartoon and sticks.

The online version of this article includes the following figure supplement(s) for figure 3:

**Figure supplement 1.** 2Fo-Fc electron density maps of LukE complexes overlaid with the corresponding models.

**Figure supplement 2.** Polder omit maps of LukE complexes overlaid with the corresponding models.

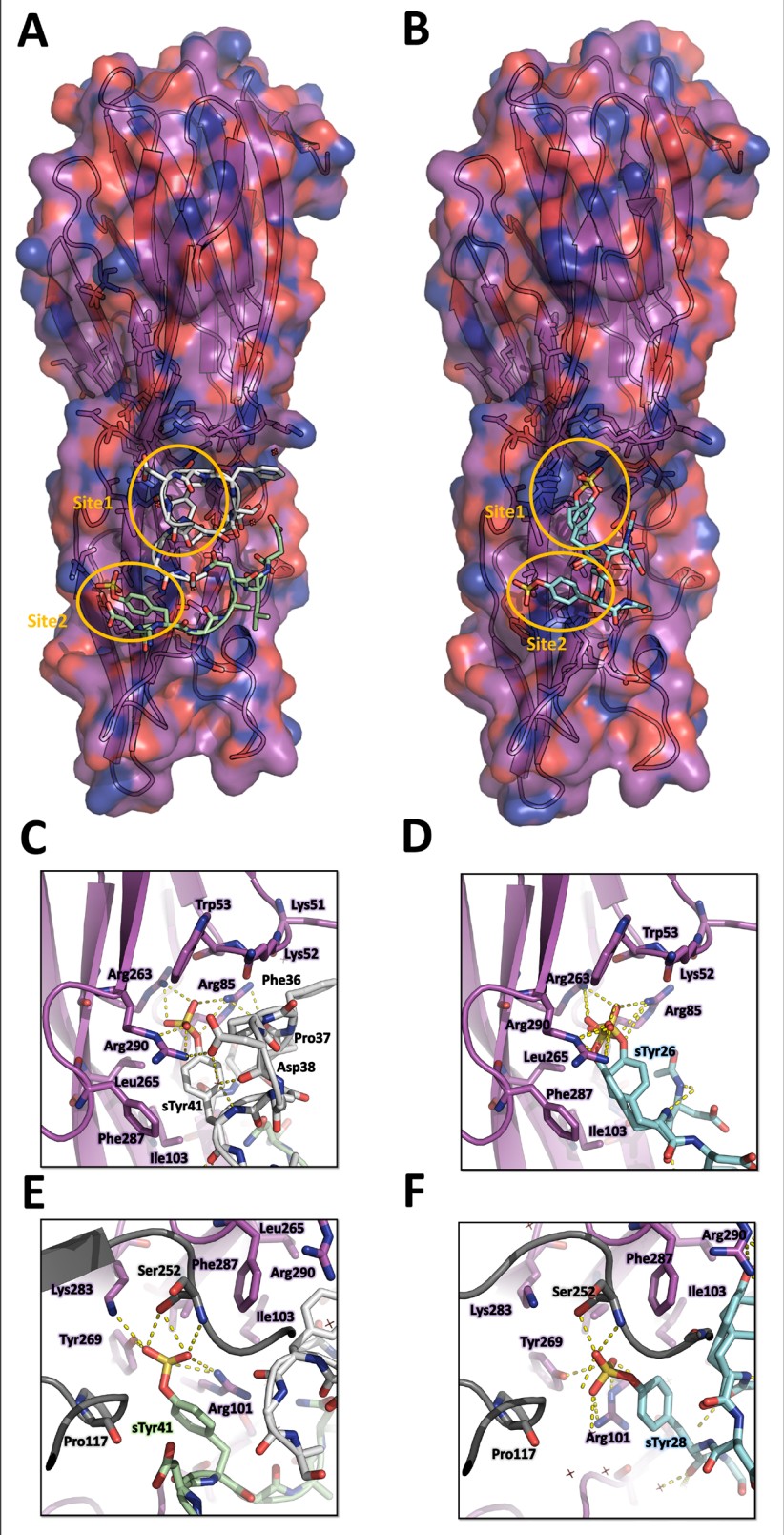

**Figure 4.** Crystal structures of LukE in complex with ACKR1 peptide [34]DSFPDGDsYGANLE[46] and CCR2 peptide [25]DsYDsYG[29]. (**A**) Overview of the asymmetric unit of the ACKR1 peptide-soaked crystal. Protein is shown in cartoon representation and colored in magenta. The two copies of the peptide with their respective sulfotyrosine residue bound to site 1 and 2 are shown in white and green cartoon and sticks representation. Surrounding protein

*Figure 4 continued on next page*

*Figure 4 continued*

sidechains are shown as magenta lines. (**B**) Overview of the asymmetric unit of the CCR2 peptide-soaked crystal. Protein is shown in cartoon representation and colored in magenta. The CCR2 peptide is shown in cyan cartoon and sticks representation. Surrounding protein sidechains are shown as magenta lines. (**C** and **D**) Close ups of the sulfotyrosine binding site 1 in the ACKR1 (**C**) and CCR2 (**D**) peptide-soaked crystals. The color code is the same as in (**A**) and (**B**) with peptide and protein sidechains shown in sticks. Polar contacts between ligand and protein are shown as yellow dashed lines. (**E** and **F**). Close ups of the sulfotyrosine binding site 2 in the ACKR1 (**E**) and CCR2 (**F**) peptide-soaked crystals. Protein residues from an interacting symmetry related molecule are shown in grey cartoon and sticks.

---

The structure of LukE bound to the CCR2 peptide $^{25}$DsYDsYG$^{29}$ is represented in *Figure 4B*. We found that sTyr26 and sTyr28 of CCR2 respectively bind to site 1 and site 2 through the same interactions observed in the pCS and ACKR1 peptide-bound structures (*Figure 4D, F*). Two alternate conformations were clearly visible for sTyr26, differing by a rotation of approximately 90° of the phenyl ring (*Figure 4C* and *Figure 3—figure supplement 1*). However, residual electron density was still present after model building and suggested the existence of a third alternate conformation of sTyr26, with its sulfate group interacting at site 2 (*Figure 3—figure supplement 1C, D*). This third alternate conformation clashes with the sulfate group of sTyr28, suggesting the existence of a minor population of molecules in which sTyr28 unbinds while sTyr26 flips its sidechain to the second site. This is consistent with the lower occupancy for site 2 versus site 1 sulfotyrosine (0.56 and 0.98), and the presence of a chain break at Asp27 in the electron density (*Figure 3—figure supplement 1*). These observations indicate that the binding of the doubly sulfated CCR2 peptide is highly dynamic within the crystal.

## Docking simulations of LukE onto ACKR1 and CCR5

In order to frame these crystallographic results into a more global context, we set out to build computational models of the full-length ACKR1-LukE and CCR5-LukE complexes. A wealth of mutagenesis data is available in the literature that provides partial information about interacting residues for the ACKR1-LukE (*Spaan et al., 2015*; *Peng et al., 2018*; *Vasquez et al., 2020*) and CCR5-LukE complexes (*Reyes-Robles et al., 2013*; *Tam et al., 2016*). The regions of LukE that are potentially implicated in receptor interaction based on previous work and on the sulfopeptide-bound structures were mapped onto the structure of LukE (*Figure 5A*), revealing a contiguous interaction surface.

The task of docking LukE onto ACKR1 and CCR5 was performed in two steps: in the first step, the toxin was docked onto the experimental structures (for CCR5) or a theoretical model (for ACKR1) without the N-terminal region of the receptor, using the information-driven docking software HADDOCK. Available mutagenesis data were encoded into Ambiguous Interaction Restraints (AIRs) by HADDOCK, providing relatively strong constraints on the orientation of the bound toxin (see Materials and methods). In the second step, the tyrosine-sulfated N-terminal region of the receptor was added to the best model of each complex, and multiple MD simulations were performed in order to simultaneously dock the N-terminal region of the receptor onto the toxin and assess the stability of the docking pose.

*Figure 5B, C* show the HADDOCK (energy) score versus iRMSD (relative to the best scoring model) for the best ACKR1-LukE and CCR5-LukE docking runs, revealing in both cases a deep energy funnel corresponding to the best cluster. The best scoring ACKR1-LukE and CCR5-LukE clusters are also the largest ones with 44 and 67 members, respectively. The best scoring pose for each complex is shown in *Figure 5D, E*, after addition of the missing N-terminal residues and prior to MD simulations. The relative orientation of LukE in each complex is somewhat different, with CCR5-bound LukE forming a more acute angle with the membrane plane compared to the ACKR1-LukE pose (*Figure 5D, E*). However, in both cases LukE loop 4 inserts into the orthosteric pocket and interacts with ACKR1 eCL3 and TMs residues, while loop 3 interacts with eCL2. Importantly, the position of TM1 and the N-terminal region of the receptor for both toxin-receptor pairs seem compatible with sulfotyrosine interactions at site 1 and/or 2 observed in crystal structures. Indeed, ACKR1 sTyr 41 is located 10 residues upstream of Cys51 that marks the beginning of TM1, while the potentially sulfated tyrosines of CCR5 (sTyr3, sTyr10, sTyr 14, and sTyr 15) are located 5–17 residues upstream of the equivalent Cys20 at the top of CCR5 TM1. This short distance between TM1 and the sulfated tyrosines, combined with the

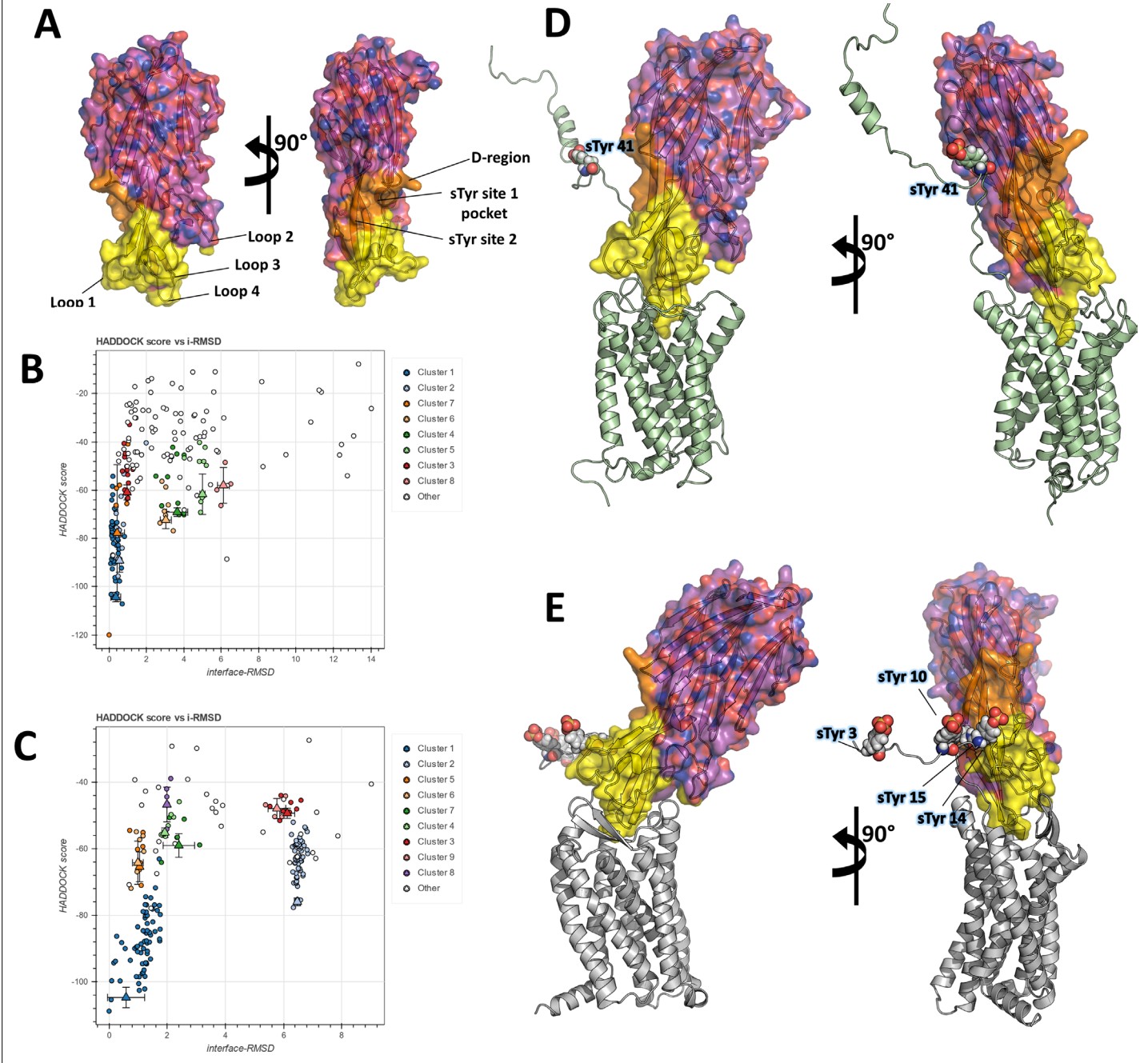

**Figure 5.** Computational docking of ACKR1-LukE and CCR5-LukE. (**A**) Mapping of the LukE residues potentially involved in receptor binding. LukE is shown in semi-transparent surface and cartoon representation and colored in magenta. Site 1 and 2 sulfotyrosine binding sites are colored in orange. Other residues that are potentially involved in receptor binding are colored in yellow based on previous mutagenesis work (*Nariya and Kamio, 1997*; *Reyes-Robles et al., 2013*; *Tam et al., 2016*; *Peng et al., 2018*; *Vasquez et al., 2020*). (**B**) and (**C**). HADDOCK score versus interface root mean square deviations (iRMSD- relative to the best scoring model) plots for the best ACKR1-LukE (**B**) and CCR5-LukE (**C**) docking runs. The best cluster identified by HADDOCK (cluster 1) is shown in blue. (**D**) and (**E**). ACKR1-LukE (**D**) and CCR5-LukE (**E**) top scoring models selected from HADDOCK simulations, after addition of the missing N-terminal region (residues 8–50 of ACKR1 and residues 1–19 of CCR5), energy minimization and MD equilibration. The receptor is shown in cartoon representation and colored in pale green (ACKR1) or light gray (CCR5). Sulfated tyrosines are shown as spheres.

known location of site 1 and 2 on LukE, provide an additional constraint on the possible orientations of the toxin, further suggesting the correctness of the identified docking poses.

In order to assess the stability of these poses and dock the N-terminal region of both receptors onto LukE, we ran multiple MD simulations starting from the equilibrated models shown in *Figure 5D*,

**Table 2. Summary of MD simulations.**

<> denotes average over simulation time. Values in parentheses correspond to standard deviation. Lines corresponding to the best MD trajectories (that were used to make *Figures 5 and 6*) are shown in bold.

| | Total simulation time (µs) | < iRMSD > (nm) | < BSA > (nm²) | < number of H-bonds> | Site 1 sulfotyrosine interaction | Site 2 sulfotyrosine interaction | |
|---|---|---|---|---|---|---|---|
| **ACKR1-LukE MD trajectory** | | | | | | | |
| **1** | **1.44** | **0.49 (0.04)** | **45.5 (6.4)** | **20.0 (4.9)** | **sTyr41** | **Asn44/Glu46** | |
| 2 | 0.84 | 0.56 (0.05) | 39.7 (7.3) | 18.9 (4.6) | sTyr41 | Glu46 | |
| 3 | 1.21 | 0.41 (0.04) | 49.5 (6.8) | 21.3 (4.8) | Not interacting | Not interacting | |
| AWH | 2.15 | 0.55 (0.07) | 41.8 (5.9) | 14.0 (4.0) | sTyr41* | Glu46* | |
| **CCR5-LukE MD trajectory** | | | | | | | |
| **1** | **1.42** | **0.37 (0.03)** | **38.9 (5.3)** | **18.5 (4.1)** | **sTyr10** | **sTyr14** | |
| 2 | 1.22 | 0.49 (0.05) | 35.4 (5.8) | 16.0 (4.5) | sTyr10 | sTyr15 | |
| 3 | 1.14 | 0.46 (0.05) | 36.2 (3.4) | 15.5 (3.7) | sTyr10 | sTyr15 | |

*Interactions observed in the global energy minimum basin of the PMF.

E (see Materials and methods). Only the functionally critical Tyr41 was sulphated in the ACKR1-LukE model, as the other sulfotyrosine in position 33 was shown to have very limited impact on LukED activity (*Spaan et al., 2015*). In the case of CCR5, no information is available about which of the potentially sulfated tyrosines are important for binding LukED, although all 4 tyrosines are heterogeneously sulfated in vivo, and sulfation is known to affect binding to other CCR5 ligands (*Farzan et al., 1999*; *Jen et al., 2009*). For these reasons, all 4 tyrosines were sulfated in our CCR5-LukE model.

## MD simulations of ACKR1-LukE docked model identify site 1 as the major sulfotyrosine binding pocket for ACKR1 N-terminal Ppptide

A summary of the information extracted from MD simulations is shown in *Table 2*. For both complexes, we monitored the number of inter-chain hydrogen bonds (H-bonds), buried surface area (BSA) and iRMSD relative to the starting model, as these parameters were recently shown to be useful in discriminating native from non-native docked models by MD (*Jandova et al., 2021*). We additionally outputted the time evolution of distances involving receptor sulfotyrosines and LukE site 1 and 2 residues (*Figure 6—figure supplement 1* and *Figure 7—figure supplement 1*), and visually inspected models for qualitative agreement with available mutagenesis data. The best models for ACKR1-LukE and CCR5-LukE were taken from the stable region of the trajectory showing the highest H-bonds, highest BSA and most stable iRMSD, while being consistent with all experimental observations (*Figure 6—figure supplement 1*, *Figure 7—figure supplement 1* and *Table 2*).

Representative MD snapshots of the best ACKR1-LukE model are shown in *Figure 6A*. The interactions between ACKR1 orthosteric pocket and LukE divergent loops initially rearranged during the first 200 ns of MD and then remained stable for the duration of the simulation, as shown by the iRMSD plateau at ~0.5 nm (*Figure 6—figure supplement 1B*). LukE Arg275 located at the tip of loop4 inserted into the orthosteric pocket, forming a stable salt bridge with ACKR1 Asp263 (*Figure 6D*). The disulfide bonded Cys51 and Cys276 interacted with a large hydrophobic surface patch of LukE involving Tyr96, Thr99, Thr271, Phe273, and Tyr 279 located in loops 1 and 4 (*Figure 6C*). Consistent with our modeling results, ACKR1 Cys51 was previously identified by mutagenesis as critical for LukED intoxication of erythrocytes (*Spaan et al., 2015*), as well as LukE loops 1, 3, and 4 (*Peng et al., 2018*; *Vasquez et al., 2020*). The N-terminal region of ACKR1 quickly attached to the toxin surface during MD (*Figure 6B*), leading to an increase in the complex BSA and number of H-bonds (*Figure 6—figure supplement 1C, D*). ACKR1 sTyr 41 bound to site 1 arginines within less than 100ns of simulation in

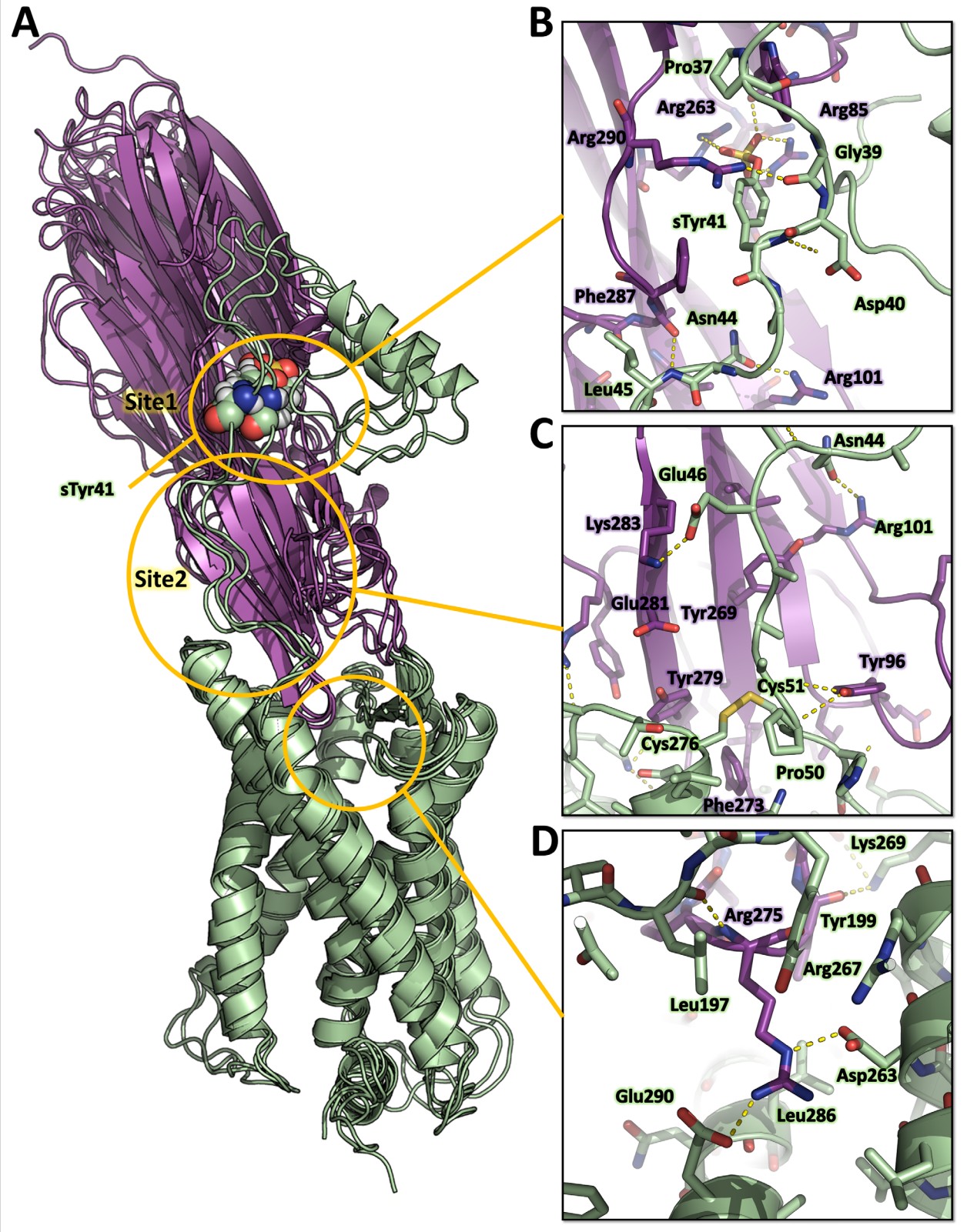

**Figure 6.** Computational model of the ACKR1-LukE complex extracted from molecular dynamics (MD) simulations. (**A**) Superimposed MD snapshots extracted from the last 300 ns of simulation (three snapshots taken at 100 ns interval). The receptor and toxin are shown in cartoon representation and colored in pale green and magenta, respectively. ACKR1 sTyr 41 is shown as spheres. (B, C and D). Close ups of the interactions observed in the proposed model at the sulfotyrosine binding site 1 (**B**), sulfotyrosine binding site 2 (**C**), and the orthosteric pocket (**D**). The MD model of ACKR1-LukE

*Figure 6 continued on next page*

*Figure 6 continued*

corresponds to the middle structure of the largest cluster in the MD trajectory. Interacting residues are displayed as sticks. The color code is the same as in (**A**).

The online version of this article includes the following video, source data, and figure supplement(s) for figure 6:

**Source data 1.** PDB file of the ACKR1-LukE MD model corresponding to the middle structure of the largest cluster in the MD trajectory.

**Figure supplement 1.** Analysis of ACKR1-LukE molecular dynamics (MD) simulations.

**Figure supplement 2.** accelerated weight histogram (AWH) molecular dynamics (MD) simulation of the ACKR1-LukE model.

**Figure 6—video 1.** Time evolution of the ACKR1-LukE model conformation and estimated PMF during AWH MD.

https://elifesciences.org/articles/72555/figures#fig6video1

2 out of 3 trajectories, while Glu46 interacted with site 2 residues Tyr269 and Lys283 (*Figure 6B* **and** *Figure 6—figure supplement 1*). The remainder of ACKR1 N-terminal region included in our model (residues 8–40) adopted various bound conformations over time in different MD trajectories, without converging to a single bound state.

Although these simulation results suggest that ACKR1 sTyr 41 interacts with LukE site 1, classical MD simulations have a known tendency to kinetically trap proteins in their closest energy minimum relative to the starting conformation (*Bernardi et al., 2015*). In order to assess the relative stability of sTyr 41 binding to site 1 versus site 2 (both of which were observed crystallographically), we performed an enhanced sampling MD simulation using the accelerated weight histogram (AWH) method (*Lindahl et al., 2014*). This method enables the calculation of potential of mean force (PMF) profiles along (a) chosen reaction coordinate(s), and functions by applying a time-dependent biasing potential that flattens free energy barriers along the reaction path. In the case of ACKR1-LukE, a two-dimensional reaction coordinate was selected to report on site 1 (sTyr 41 – Arg263 distance) and site 2 (sTyr41 – Tyr269 distance) interactions. After extensive sampling of the reaction coordinate (*Figure 6—figure supplement 2A* and *Figure 6—video 1*), the final PMF profile extracted from the AWH MD simulation at t ~2.15 μs showed a deep energy well with a sTyr 41 – Arg263 distance of ~0.8 nm and sTyr41 – Tyr269 distance of ~1.2–1.6 nm (*Figure 6—figure supplement 2B*), indicating a clear preference for binding site 1 (*Figure 6—figure supplement 2C*).

## MD simulations of the CCR5-LukE docked model indicate that CCR5 N-terminal peptide recognizes both site 1 and 2

Representative MD snapshots of the best CCR5-LukE model are shown in *Figure 7A*. The toxin-receptor relative orientation and interations between LukE loops and CCR5 orthosteric pocket remained mostly stable during MD with iRMSD values of 0.4–0.5 nm (*Figure 7—figure supplement 1A, B*). The observed interactions were consistent with information available in the literature (*Reyes-Robles et al., 2013*; *Tam et al., 2016*) and involved extensive contacts between CCR5 eCL2 and LukE loops 2/3 (*Figure 7E*), and the upper part of TM 5, 6, and 7 with loop 4 (*Figure 7D*). After initial rearrangements, LukE Arg275 from loop 4 formed stable ionic interactions with CCR5 Asp276 and Glu283 located in the upper part of TM7 (*Figure 7D*). Similar to the ACKR1-Luke model, the N-terminal region of CCR5 rapidly docked itself onto the toxin surface in a process most likely driven by long range electrostatics, creating interactions with LukE loop 1 and with the sulfotyrosine binding sites (*Figure 7B, C*). In all 3 MD trajectories, CCR5 sTyr 10 bound to LukE sulfotyrosine site 1 after ~100–400 ns of simulation time (*Figure 7C* and *Figure 7—figure supplement 1E*). Binding of sTyr 14 or 15 to sulfotyrosine site 2 were also observed in all trajectories, however this interaction was more dynamic, possibly because both residues were sulfated and able to interact with LukE site 2 residues (*Figure 7B* and *Figure 7—figure supplement 1F, G*). The first N-terminal residues of CCR5 (residues 1–9) adopted various conformations in different trajectories without converging to a stable bound state.

These results were further confirmed by running an AWH MD simulation of CCR5-LukE using sTyr 10 – Arg263 and sTyr10 – Tyr269 distances as a two-dimensional reaction coordinate, in order to validate the interaction of sTyr 10 with site 1 that was observed in classical MD. After extensive sampling of the reaction coordinate (*Figure 7—figure supplement 2A* and *Figure 7—video 1*), the final PMF profile extracted from the AWH MD simulation at t ~ 2.05 μs showed a deep energy well with a sTyr 10 – Arg263 distance of ~0.8 nm and sTyr10 – Tyr269 distance of ~1.4–1.7 nm (*Figure 7—figure*

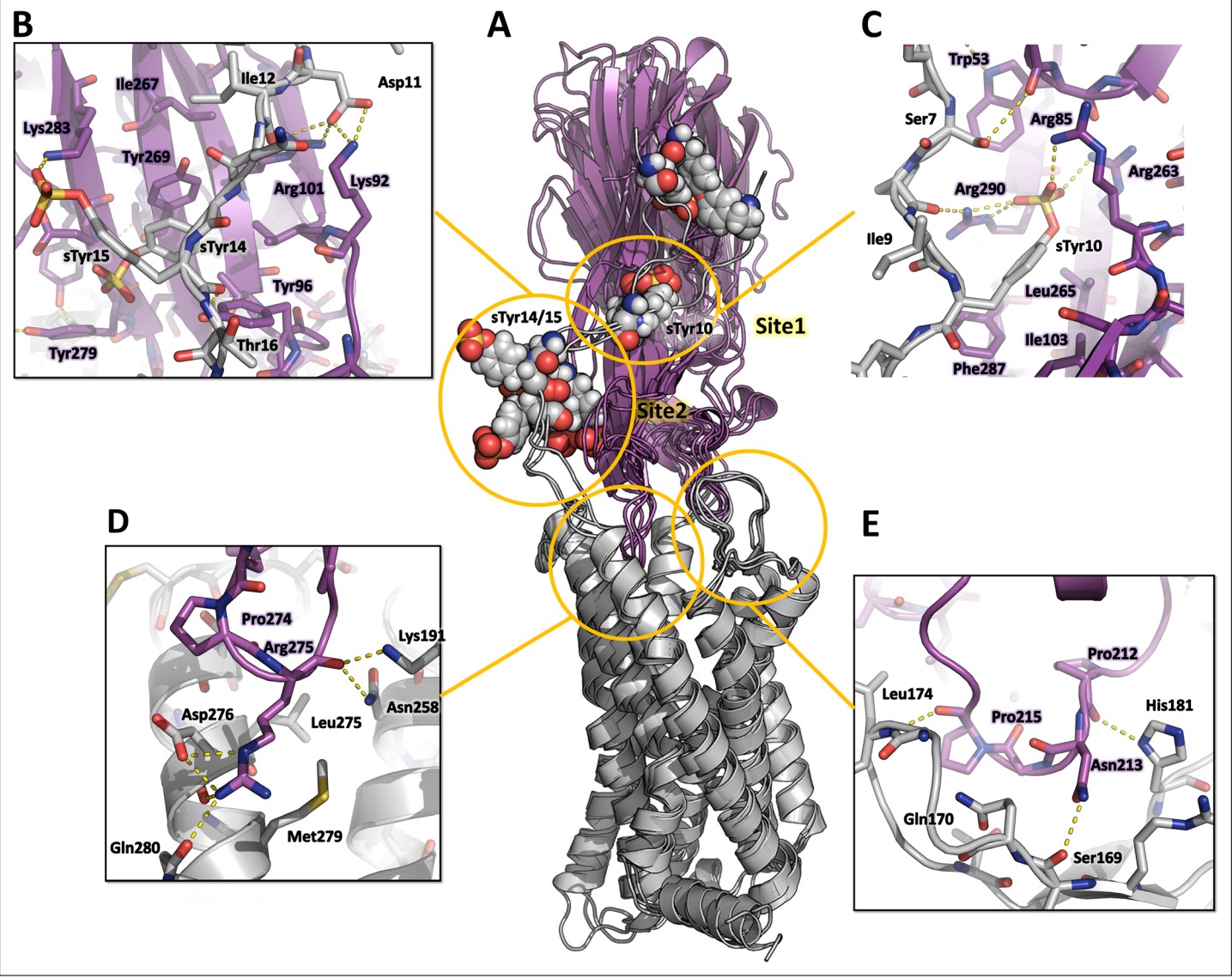

**Figure 7.** Computational model of the CCR5-LukE complex extracted from molecular dynamics (MD) simulations. (**A**) Superimposed MD snapshots extracted from the last 300ns of simulation (3 snapshots taken at 100 ns interval). The receptor and toxin are shown in cartoon representation and colored in light gray and magenta, respectively. CCR5 sTyr residues are shown as spheres. (**B–E**). Close ups of the interactions observed in the proposed model at the sulfotyrosine binding site 2 (**B**), sulfotyrosine binding site 1 (**C**), orthosteric pocket (**D**) and CCR5 eCL2/LukE loop3 region (**E**). The MD model of CCR5-LukE corresponds to the middle structure of the largest cluster in the MD trajectory. Interacting residues are displayed as sticks. The color code is the same as in (**A**).

The online version of this article includes the following video, source data, and figure supplement(s) for figure 7:

**Source data 1.** PDB file of the CCR5-LukE MD model corresponding to the middle structure of the largest cluster in the MD trajectory.

**Figure supplement 1.** Analysis of CCR5-LukE molecular dynamics (MD) simulations.

**Figure supplement 2.** Accelerated weight histogram (AWH) molecular dynamics (MD) simulation of the CCR5-LukE model.

**Figure 7—video 1.** Time evolution of the CCR5-LukE model conformation and estimated PMF during AWH MD.
https://elifesciences.org/articles/72555/figures#fig7video1

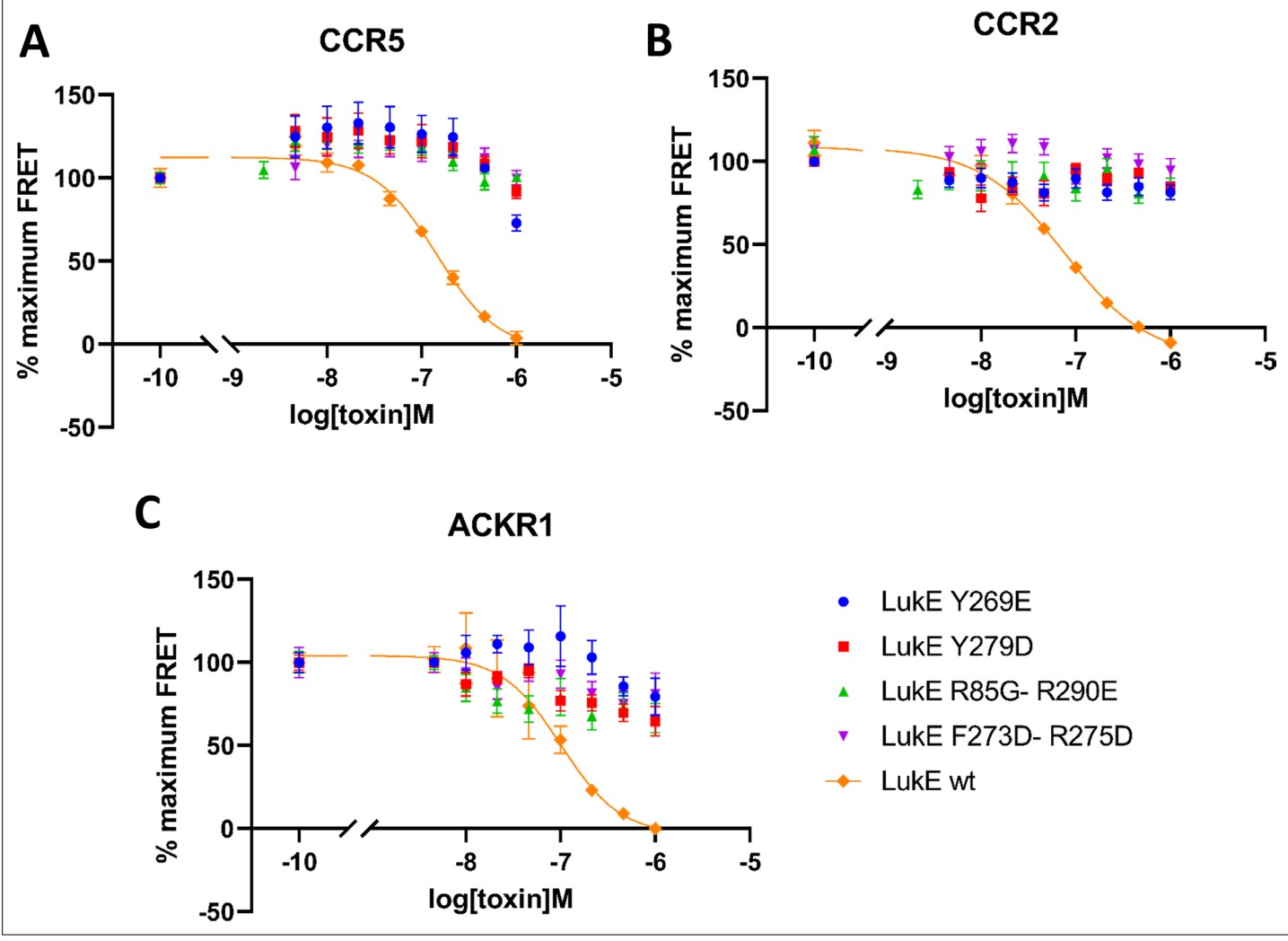

**Figure 8.** Effects of R85G + R290 E, Y269D, Y279D, and F273D + R275 D mutations on the binding of LukE in live cells as determined by competition time-resolved fluorescence energy transfer (TR-FRET). 5 nM tracer ligands, CCL5-d2 for CCR5 (**A**) and ACKR1 (**C**) and CCL2-d2 for CCR2 (**B**), were used to determine TR-FRET at their respective receptors in the presence of LukE mutants or wild-type LukE. IC50 values were undetermined for LukE mutants due to loss of binding. Data shown is mean ± SEM of three independent experiments performed in triplicate.

The online version of this article includes the following figure supplement(s) for figure 8:

**Figure supplement 1.** Environment of the mutated LukE residues in the ACKR1-LukE model.

**Figure supplement 2.** Environment of the mutated LukE residues in the CCR5-LukE model.

_supplement 2B_), indicating a clear preference of sTyr10 for binding site 1 (_Figure 7—figure supplement 2C_). In addition, the models corresponding to this main energy basin displayed interaction of sTyr14 with site 2.

## LukE mutants lose their ability to inhibit chemokine binding to CCR5, ACKR1, and CCR2

In order to validate experimentally the interactions identified in the computational models of ACKR1-LukE and CCR5-LukE, we designed four mutants of LukE to disrupt the interactions with CCR5/ACKR1 at the orthosteric pocket and sulfotyrosine binding site 1 and 2, which we evaluated using the TR-FRET assay (_Figure 8_). The environment of each mutated residues is shown in figure S8 and S9 for the ACKR1-LukE and CCR5-LukE models, respectively. The R85G-R290E mutant neutralizes sulfotyrosine binding site 1 and favors intramolecular side chain interactions between R263 and mutated E290 (_Figure 8—figure supplement 1A_ and _Figure 8—figure supplement 2A_). The Y269E mutant disrupts

sulfotyrosine binding site 2. The mutation favors intramolecular side chain interactions between R101/K283 and mutated E269, destabilizing the intermolecular interactions with CCR5 sTyr14 or ACKR1 Asn44/Glu46 (*Figure 8—figure supplement 1B* and *Figure 8—figure supplement 2B*). The Y279D mutant disrupts the interaction with Glu262/Asn267 in the CCR5-LukE model, and with Cys51-Cys276 in the ACKR1-LukE model (*Figure 8—figure supplement 1C* and *Figure 8—figure supplement 2C*). The F273D-R275D disrupts multiple interactions at the orthosteric pocket site, including R275-D276/E283 in CCR5-LukE and R275-D263/E290 in ACKR1-LukE. In addition, in ACKR1-LukE, F273 is part of an hydrophobic cluster that includes Y279/F273 of LukE and Cys51-Cys276 of ACKR1 (*Figure 8—figure supplement 1D* and *Figure 8—figure supplement 2D*). As expected, the activity of the mutants in the TR-FRET binding assay was abolished or strongly decreased for all receptors (*Figure 8*), providing an additional validation of our models.

## Discussion

The present study provides an in-depth characterization of the structure of monomeric LukE and of its ability to bind the chemokine receptors ACKR1, CCR2 and CCR5. TR-FRET competition assays indicated that LukE can displace bound CCL2/CCL5 from their cognate receptors, confirming significant overlap between the chemokine and toxin binding surfaces on the receptors. Surprisingly, LukE competed with CCL2 for CCR2 binding with a slightly better IC50 than for the other receptors, suggesting a strong interaction between CCR2 and LukE. This indicates that even though LukED is unable to lyse CCR2-expressing cells (*Tam et al., 2016*), LukE can still prevent CCR2 signalling. Such competitive inhibition might exist for other chemokine receptors bearing sulfotyrosines in their N-terminus, and could play a role in SA infections by modulating inflammatory responses.

Based on the information available in the literature, it appeared that sulfotyrosine recognition may constitute a common feature of leukotoxin–GPCR interactions, which may in part explain their broad receptor specificity. We thus sought to obtain high resolution structural information regarding this type of interactions. The crystal structure of LukE in complex with p-cresyl sulfate identified 3 potential sulfotyrosine binding sites, 2 of which were also observed in the LukE structures in complex with peptides derived from the N-terminal region of ACKR1 and CCR2. We then used protein-protein docking and MD simulations to propose models of the ACKR1-LukE and CCR5-LukE complexes that are consistent with the crystallographic data and with previous mutagenesis data (*Reyes-Robles et al., 2013*; *Spaan et al., 2015*; *Tam et al., 2016*; *Peng et al., 2018*; *Vasquez et al., 2020*). Our work demonstrates that site 1 is used to bind ACKR1 sTyr 41, CCR2 sTyr 26, and CCR5 sTyr 10, while site 2 can accommodate an additional downstream sulfotyrosine such as CCR5 sTyr 14 or 15, or CCR2 sTyr 28.

### Conservation of the sulfotyrosine binding sites among GPCR-binding leukotoxins

Site 1 is widely conserved across leukotoxin S components, and the three arginines (Arg 85, 263, and 290 of LukE) coordinating the tyrosine sulfate are strictly conserved in HlgA, HlgC and lukS-PV (*Figure 9* BDEG). The three hydrophobic residues that are in direct interaction with the sulfotyrosine phenyl ring (Ile103, Leu265, Phe287 of LukE) show more variability, with Leu265 mutated to Met in HlgA, and both Leu265 and Ile103 changed to Met and Arg in HlgC and LukS-PV. This high level of conservation strongly suggests that all four toxins use site 1 to recognize sulfated tyrosines in their respective receptors. These receptors contain sulfotyrosines within a YXY (CXCR2, CCR2), YXXY (C5aR1, C5aR2), or YXXXY (CCR5) motif, except for ACKR1 and CXCR1 that possess isolated sulfotyrosine residue(s) (*Figure 9A*).

Consistent with these observations, the post-translational modification pathways involved in the sulfation of the leukocidin receptors were recently found to impact on HlgAB, HlgCB, LukED, and PVL induced cytotoxicity, most likely through reduced S component binding (*Tromp et al., 2020*). In addition, ACKR1 Tyr41 to Ala mutation was shown to nearly abolish LukED- and HlgAB-induced hemolysis in HEK cells (*Spaan et al., 2015*), while ACKR1 Tyr41 to Phe mutation caused a 10 fold increase in HlgAB EC50 required to dissociate ACKR1-bound Gαi subunit, as measured by BRET (*Grison et al., 2021*). LukS-PV binding to C5aR1 was shown to require tyrosine sulfation as well, with a measured affinity of 127 nM (SD ±17 nM) between lukS-PV and the receptor N-terminal peptide by ITC (*Spaan*

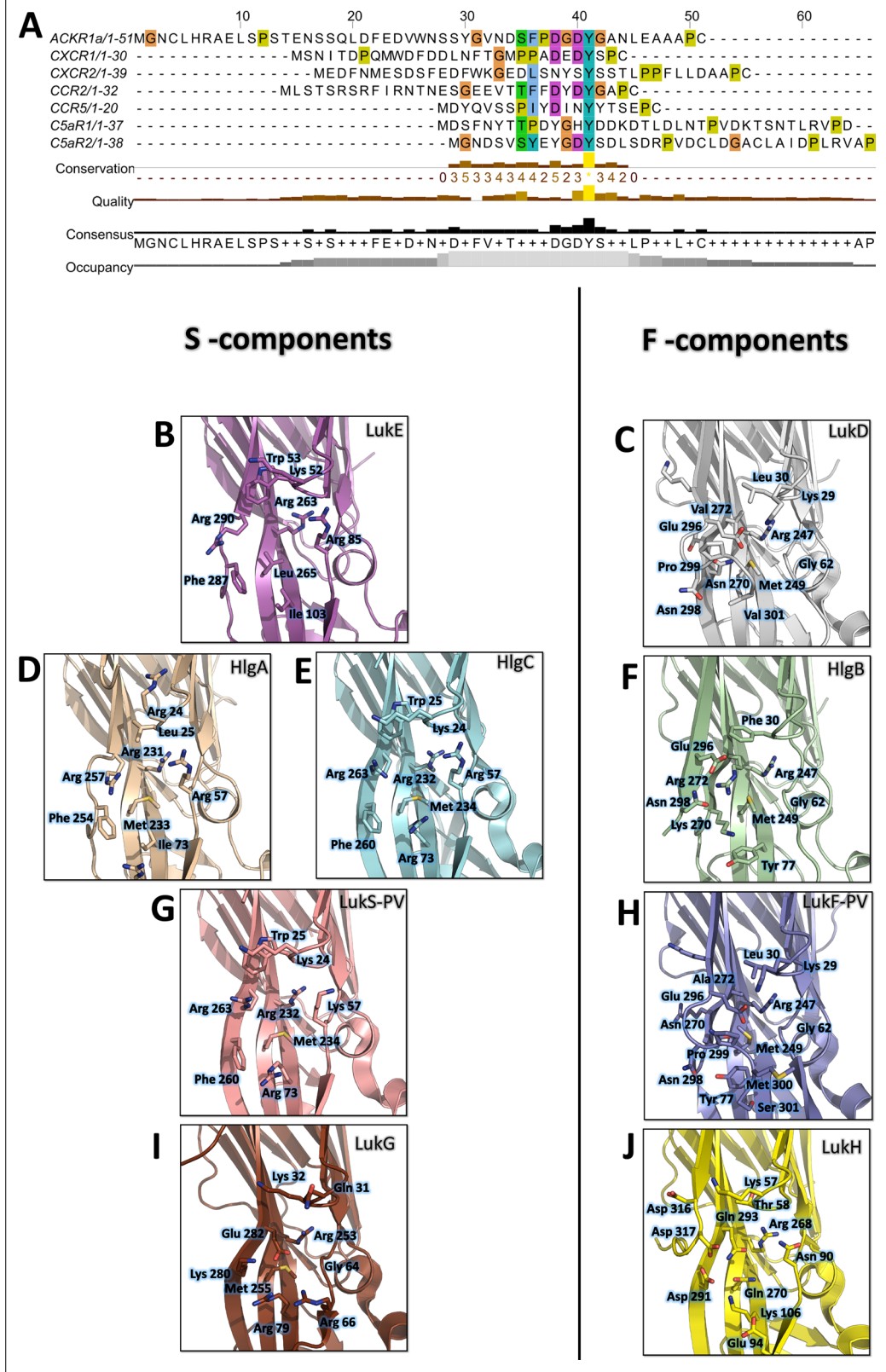

**Figure 9.** Conservation of the sulfotyrosine sites in the N-terminal region of leukotoxins receptors and structural alignment of *Staphylococcus aureus* bicomponent pore forming toxins in the sulfotyrosine binding region (site 1). (**A**) Manual sequence alignment of the sulfotyrosine motifs in the N-terminal region of the bicomponent pore forming toxin receptors. (**B–J**) Comparison of sulfotyrosine site 1 residues in SA bi-component leukotoxin x-ray

*Figure 9 continued on next page*

*Figure 9 continued*

structures. (**B**) LukE apo1. (**C**) LukD (PDB ID 6U2S, *Liu et al., 2020*). (**D**) HlgA (PDB ID 2QK7, *Roblin et al., 2008*). (**E**) HlgC (PDB ID 4P1X, *Yamashita et al., 2014*). (**F**) HlgB (PDB ID 3LKF, *Olson et al., 1999*). (**G**) LukS-PV (PDB ID 1T5R, *Guillet et al., 2004*). (**H**) LukF-PV (PDB ID 1PVL, *Pédelacq et al., 1999*). (**I** and **J**) LukG and lukH (PDB ID 5K59, *Badarau et al., 2016*).

*et al., 2013*). Interestingly, a bound sulfate ion is also visible at LukS-PV site 1 in the PVL crystal structure reported in *Liu et al., 2020* (PDB CODE 6U3Y). Taken together, these data indicate that the sulfotyrosine - site 1 interactions observed in this study play a critical role in receptor recognition by S-components.

Site 2 is only conserved in LukE and HlgA, and likely confers to these toxins the ability to bind the second sTyr of the motif in the N-termini of CCR2, CCR5, and CXCR2. Divergence in site 2 residues for hlgC and LukS-PV indicates that these toxins might use a different set of residues to bind the second Y in C5aR1 and C5aR2 YXXY motif.

Contrary to S components, F-components show very poor conservation of site 1, except perhaps for HlgB, in which 3 out of 6 sulfotyrosine-interacting residues are conserved with HlgA/C and lukS-PV (*Figure 9B-J*). Given the recently described interaction between ACKR1 and HlgB (*Grison et al., 2021*), it is tempting to speculate that these residues might play a role in receptor recognition, although further experiments will be necessary to (in)validate this hypothesis. Intriguingly, the Arg residue sitting at the bottom of site 1 pocket (LukE Arg263) is conserved in all toxins, including the more distantly related lukGH. Nevertheless, F components harbour a C-terminal extension of their last β strand with an exposed acidic residue (for example LukD Glu296) that can neutralize the positively charged Arg (*Figure 9C, F, H, J*).

## Implications of the proposed ACKR1- and CCR5-LukE models for receptor binding, conformational changes and subsequent pore formation

In this study, we used computational docking and MD simulations to produce plausible models of the ACKR1- and CCR5-Luke complexes that are consistent with all available experimental data. During most MD simulations, we observed fast binding of ACKR1 sTyr 41 or CCR5 sTyr 10 to LukE Arg263 of site 1 (*Figure 6—figure supplement 1E* and *Figure 7—figure supplement 1E*), with an inter-residue distance going from ~3 nm to ~0.2 nm within a few hundred nanoseconds or less. Such fast binding indicates that the clustering of positive charges in site 1 generates large electrostatic forces leading to long-range attraction of the negatively charged N-terminal region of chemokine receptors. This, in turn, suggests that sulfotyrosine recognition through site 1 may be the first step in receptor recognition, followed by binding of the divergent regions/loops to the receptor's orthosteric pocket.

Although our models offer a glimpse of how LukE might bind to ACKR1 and CCR5 orthosteric pocket, it should be emphasized that computational models are error-prone and thus might be inaccurate. Moreover, the timescale of the simulations used is limited and unable to account for potential large-scale receptor conformational changes that toxin binding may induce. For the same reason, it is not currently possible to determine from the data whether LukE- receptor binding might allosterically induce STEM release in LukE, which would be a necessary step to enable LukD binding and subsequent pore assembly without inter-protomeric steric clashes (*Yamashita et al., 2011*; *Yamashita et al., 2014*; *Liu et al., 2020*). Thus, the proposed models might only represent initial encounter complexes that are subject to further conformational changes.

With these limitations in mind, a comparison of the proposed models of ACKR1-LukE and CCR5-LukE with the available (pre)pore structures of HlgAB, PVL, and lukGH in complex with human CD11b I-domain yields a few interesting observations:

First, toxin-receptor and toxin-toxin interactions are mediated by distinct, non-overlapping surfaces, as was also observed for the lukGH - CD11b I-domain complex (*Figure 10A, B*). This indicates that the initial step of receptor recognition is compatible with subsequent toxin oligomerization.

Second, the depth of insertion of LukE into the receptors' orthosteric pocket is insufficient to allow the pore's β-barrel to fully span the membrane. Again, this might be due to model inaccuracies and/or conformational changes of the receptors unaccounted for in our modeling procedure. An alternative possibility is that receptors might serve as an assembly platform for the prepore structure

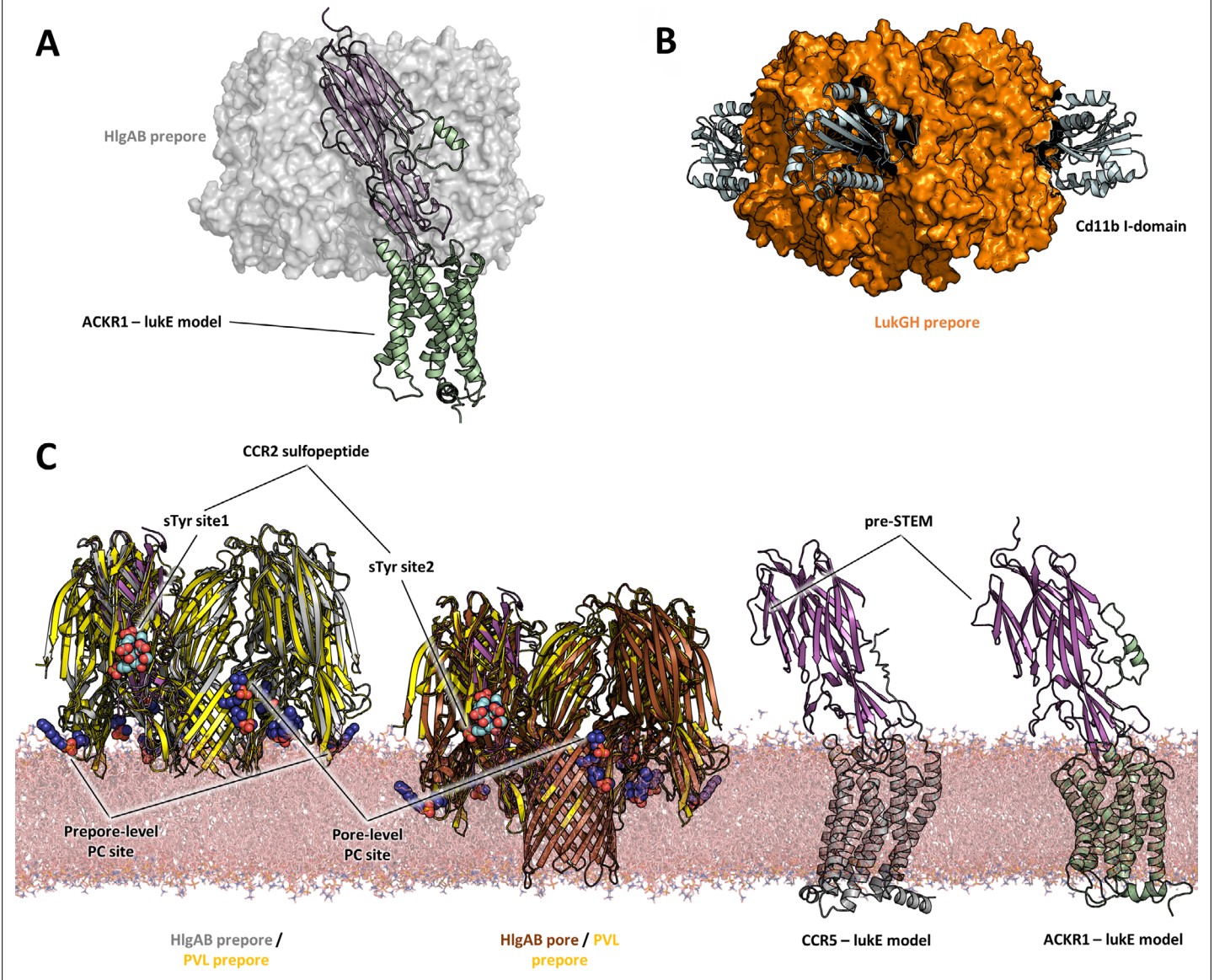

**Figure 10.** Comparisons of LukE-CCR2 peptide x-ray structure, ACKR1-LukE and CCR5-LukE models with existing x-ray structures. (**A**) Structural alignment of LukE from the ACKR1-LukE model with the HlgAB prepore structure (PDB ID 4P1Y), illustrating that the N-terminal region of ACKR1 binds to a surface located on the outer part of the prepore. The HlgAB prepore is shown as a semitransparent surface and the ACKR1-LukE model as cartoon. (**C**) X-ray structure of LukGH bound to human CD11b I-domain. The LukGH prepore is shown as a semitransparent surface and the CD11b I-domain as cartoon. (**C**) The HlgAB pore and prepore structures (PDB IDs 3B07 and 4P1Y) are shown in their membrane context using a POPC bilayer extracted from MDS simulations. On the left, the HlgAB pore and prepore structures are superimposed to the PVL prepore (PDB ID 6U3Y) bound with fos-choline-14 (shown as blue spheres). One of the lukF-PV/HlgB protomer is omitted for clarity. The structural alignment highlights the presence of lipid headgroup binding sites on lukF-PV that correspond to different membrane insertion depths, which we interpreted as pore and prepore membrane levels. The LukE-CCR2 peptide structure is also superimposed onto the pore and prepore structures to indicate the location of the bound sulfotyrosines (shown as cyan spheres). In the case of a fully formed pore, sTyr sulfate at site 2 is located at the same level as the PC headgroups. On the right, the ACKR1-LukE and CCR5-LukE models are shown in their membrane context, and oriented to visualize the acute angle between LukE and the membrane.

but be released from the interaction prior to full pore formation. In support of this hypothesis, the crystal structure of PVL (pre)pore bound to fos-choline-14 has shown binding of 3 PC headgroups in different regions of LukF-PV corresponding to different membrane insertion depths, with the highest one compatible with the full membrane spanning pore (*Figure 10*). When compared with the sulfotyrosine-bound crystal structures reported here, this would imply that site 2 is at membrane level, which would be incompatible with receptor binding. On the contrary, the 2 other PC binding

sites, which have also been identified in lukD structures, are located in regions compatible with the level of membrane insertion implied by our ACKR1-LukE and CCR5-LukE models. Importantly, these observations are also consistent with the study by Haapasalo et al. which showed that C5a receptors dissociate from lukSF-PV upon pore formation, enabling renewed ligand binding and pore formation (*Haapasalo et al., 2019*).

Finally, our ACKR1-LukE and CCR5-LukE models suggest that the bound toxin could make an acute angle with the membrane that may bring the preSTEM region closer to the lipid headgroups (*Figure 10*), possibly playing some role in STEM release.

In summary, we have elucidated the structural basis for recognition of sulfotyrosine motifs located in the N-terminal region of leukotoxin receptors and used this data to propose models of the ACKR1-LukE and CCR5-LukE encounter complexes. We then used these models to design LukE mutants that disrupt key interactions within the modelled complexes, and showed that the mutants lost their receptor binding activity in the TR-FRET assay. Although the combination of crystallographic and mutagenesis data provide substantial validation for these models, they might still suffer from inaccuracies and may be further improved by additional interface mapping through biophysical approaches (such as HDX-MS or NMR for instance). They also constitute a starting point to design strategies to obtain high-resolution structural data, as, so far, the purified complexes were reluctant to LCP crystallography and cryoEM analyses. However, in the absence of high resolution cryoEM or x-ray structures of full-length GPCR-toxin complexes, these models help refine our understanding of the mechanisms driving receptor recognition and subsequent pore formation by SA leukotoxins. Sulfotyrosine site 1 additionally appears to have a unique architecture, which is distinct from the recognition motifs used by chemokines. The binding site is widely conserved across S-components, suggesting that this pocket might be a promising target for structure-based design of inhibitors preventing receptor recognition. Such inhibitors might be active against a broad range of SA strains expressing different leukotoxins.

# Materials and methods

## Protein constructs and reagents

The synthetic gene of *S. aureus* LukEv (UniprotKB-Q2FXB0) residue 12–311 (the product of which was successfully expressed and crystallized in *Nocadello et al., 2016*) was subcloned into popinE vector (OPPF-UK), resulting in a construct bearing a part of the N-terminal signal sequence ($^{12}$SVGLIAPLASPIQESRA$^{28}$) and a C-terminal K(H)$_6$-tag. For cell-based assays, we used SNAP-tagged ACKR1, CCR2, and CCR5 constructs. The synthetic genes of each receptor were subcloned into pcDNA3.1-SNAP vector in frame with the C-terminus of SNAP tag. d2-labeled CCL2 and CCL5 were purchased from Almac. P-cresol sulfate was purchased from Sigma-Aldrich (SMB00936). R85G + R290E, Y269D, Y279D, and F273D + R275D mutants of LukE were purchased from GeneCust.

## Expression and purification of LukE

C-terminally (His)$_6$-tagged LukE was expressed in BL21 (DE3) *Escherichia coli* cells (NEB). Transformed cells were grown at 37°C in Terrific broth supplemented with 100 µg/ml ampicillin to a density of OD600 = 0.6. Expression was then induced overnight at 18°C by addition of 0.5 mM IPTG. Cells were harvested by centrifugation (3,000 rpm) and cell pellets were stored at − 80°C until purification. After thawing the frozen cell pellets, cells were lysed by sonication in a lysis buffer consisting of 10 mM Tris (pH 8) and 150 mM NaCl. Lysed cells were centrifuged (16,000 rpm) and the supernatant was loaded onto a nickel NTA Agarose resin. The resin was washed with 20 CV of wash buffer 1 consisting of 50 mM HEPES (pH 7.5) and 1 M NaCl, and with 10 CV of wash buffer 2 consisting of 50 mM HEPES (pH 7.5), 150 mM NaCl and 40 mM imidazole. Bound toxin was eluted with wash buffer 2 supplemented with 200 mM imidazole. The eluted proteins were concentrated using 30 kDa spin filters (Millipore) and further purified by size exclusion chromatography on a Superdex 200 Increase 10/300 column (GE Healthcare) in 50 mM HEPES (pH 7.5) and 150 mM NaCl. Monodisperse elution fractions were pooled and further concentrated prior to crystallization, SAXS and/or cell-based assays.

## Native mass spectrometry

Prior to MS analysis, proteins were buffer exchanged into 200 mM ammonium acetate buffer pH 7.4 (Sigma) using Bio-Spin microcentrifuge columns (Bio-Rad Laboratories). Intact MS spectra were recorded on a Synapt G2-Si HDMS instrument (Waters Corporation) modified for high mass analysis and operated in ToF mode. Samples were introduced into the ion source using borosilicate emitters (Thermo Fisher Scientific). Optimized instrument parameters were as follows: capillary voltage 1.4 kV, sampling cone voltage 80 V, offset voltage 80 V, transfer collision voltage 25 V and argon flow rate 5 ml/min. Collision voltage in the trap was optimized between 50 and 110 V. Data was processed using MassLynx v.4.2 (Waters).

## Competitive binding assays by TR-FRET

HEK293T cells (ATCC CCRL-3216, human embryonic kidney) were grown in Dulbecco's Modified Eagle Medium supplemented with 10% FBS and 1% Penicillin/streptomycin at 37°C with 5% $CO_2$. Transfections were performed using lipofectamine 2000 (Life technology) in polyornithine coated white 96-well, opaque-bottom plates. For CCR5 and ACKR1 constructs, cells were seeded into six-well plates at a density of 300,000 cells/well before being transfected overnight with 100 ng/well of SNAP-CCR5 or SNAP-ACKR1. An empty PRK vector was added to obtain a total of 3 µg of DNA/well. After 24 hr, cells were further seeded into 96-well white, opaque plates at a density of 30,000 cells/well. CCR2 was transfected directly into 96 well plates by adding 2 ng of SNAP-CCR2 to 30,000 cells/well and empty PRK vector was added for a total 200 ng/well of DNA. 24 hr after seeding into 96-well plates, cells were washed with TagLite buffer (Cisbio, Codolet, France) and the receptors were labeled with SNAP-Lumi4-Tb (100 nM, Cisbio, Codolet, France) for 1 h at room temperature. Cells were washed four times with TagLite and then treated with increasing concentrations of LukE. 5 µM of a fluorescent tracer was then added to wells and incubated for 4 hr at 4°C. CCL5-d2 was used as the tracer for CCR5 and ACKR1 and CCL2-d2 was used for CCR2. Fluorescence readouts were osberved on a Pherastar plate reader (BMG Labtech) or Spark 20 M reader (Tecan): samples were illuminated at 337 nm and fluorescence was acquired at 620 nm (donor) and 665 nm (acceptor). The ratio of the signals (665/620) was calculated and plotted against toxin concentration. Dose-response curves were generated using GraphPad Prism 6 (GraphPad Software, Inc, San Diego, CA).

## Small angle x-ray scattering

Small angle x-ray scattering measurements of LukE at 1, 2, and 4 mg/ml in 50 mM HEPES (pH 7.5) and 150 mM NaCl were performed at the BioSAXS beamline (BM29) of the European Synchrotron Radiation Facility (ESRF). Data was collected at 20°C, a wavelength of 0.0995 nm and a sample-to-detector distance of 1 m. The scattering from the buffer alone was measured before and after each sample measurement and was used for background subtraction with PRIMUS from the ATSAS package (*Franke et al., 2017*).

## Molecular dynamics and ensemble optimization of LukE

Starting coordinates for LukE were taken from the Apo2 structure. Missing terminal residues (residues 24–28 and the C-terminal polyhistidine tag) were added in extended conformations in Coot (*Emsley and Cowtan, 2004*). MD simulations were performed using GROMACS 2020 (*Hess et al., 2008*) and either the AMBER99SB-ILDN* force field (*Best and Hummer, 2009*; *Lindorff-Larsen et al., 2010*), or the AMBER99SBWS force field (*Best et al., 2014*). The rationale for using 2 force fields in the MD simulations of LukE was to generate more diverse conformations of the protein. The AMBER99SB-ILDN* force field tends to favor collapsed states for the intrinsically disordered regions (signal peptide and polyhistidine tag in this case), while the AMBER99SBWS force field samples more effectively the extended conformations. At the beginning of each simulation, the protein was immersed in a box of SPC/E or TIP4P water, with a minimum distance of 1.0 nm between protein atoms and the edges of the box. 150 mM NaCl were added using genion. Long range electrostatics were treated with the particle-mesh Ewald summation (*Essmann et al., 1995*). Bond lengths were constrained using the P-LINCS algorithm (*Hess, 2008*). The integration time step was 5 fs. The v-rescale thermostat (*Bussi et al., 2007*) and the Parrinello–Rahman barostat were used to maintain a temperature of 300 K and a pressure of 1 atm. Each system was energy minimized using 1,000 steps of steepest descent and equilibrated for 200 ps with restrained protein heavy atoms. A single 500 ns production simulation

was run using each forcefield. RMSF and radius of gyration were calculated using GROMACS routines. Snapshots were extracted every 500 ps, resulting in a pool of 2,000 models. Theoretical SAXS patterns were calculated with the program CRYSOL (*Svergun et al., 1995*) and ensemble fitting was performed with GAJOE (*Bernadó et al., 2007*).

## Peptide synthesis

ACKR1 $^{34}$**DSFPDGDsYGANLE**$^{46}$ *and CCR2* $^{25}$**DsYDsYG**$^{29}$ sulfopeptides used in crystal soaking experiments were synthesized using solid-phase synthesis (SPPS) methodology with Fmoc/tBu protocol and all amino acids were incorporated into the sequence by the building block approach. Those synthesis was carried out by hand using the syringe technique on Fmoc-Rink Amide Resin (Amphisphere 40 RAM) with a substitution value of 0.35 mmol/g. Side chain protecting groups that were used in this synthesis were *t*Bu for Asp,Glu, and Ser; Trt for Asn and Fmoc-Tyr(SO$_3$Na)-OH was synthesized separately in solution. Fmoc groups for *Nα*-protection were cleaved by treatment with 20% piperidine in DMF in two times. After each deprotection of Fmoc group or coupling step, resin was washed as follow: 3 times DMF, 2 times MeOH, 2 times DMF and 2 times DCM.

The appropriate amino acid (3 eq.) was dissolved in DMF in presence of HATU (2.9 eq.) and DIEA (4.5 eq.) for 5 min. This solution was then added to the resin and leaved under agitation at room temperature. Effectiveness of the reaction was monitored by disappearance of primary amino group at the N terminus of the growing peptide. This was checked by a negative Kaiser test or Chloranil test for proline residue. Durations of coupling steps ranged from 2 hr to overnight. After the last deprotection of Fmoc group, acetylation on N-terminal group was performed twice for 15 min with a solution of DMF/Ac$_2$O (4:1; v/v) and 1% of DIEA.

Then after a wash step, the side-chain protection groups were removed and the peptides were cleaved from the resin using a mixture of 90% TFA and 10% H20 for 2 hr at 0°C ($^{25}$**DsYDsYG**$^{29}$) or 6 hr at 0°C ($^{34}$**DSFPDGDsYGANLE**$^{46}$). After filtration of the resin and concentration of filtrat at room temperature, crude peptide is precipitate by dropwise in diethylether at 0°C. Crude peptide is then dissolve in a mix of water and acetonitrile. pH was adjusted above 9 with a solution of NH$_4$OH aq 10% before to be lyophilisate.

Crude peptide was purified by HPLC in formiate buffer (pH 4.5) and LC-MS analyses were performed by Electrospray negative mode.

## X-ray crystallography

Crystallization was carried out by vapor diffusion using a Cartesian Technologies pipetting system (*Walter et al., 2005*). LukE was concentrated to about 12 mg/ml (390 µM) in 25 mM HEPES pH 7.5 and 75 mM NaCl, and in the presence of 6 molar equivalents of the ACKR1 sulfopeptide. The protein crystallized after ~3–7 days at 20°C in 0.1 M imidazole.HCl pH 8.0, 30% (w/v) MPD, 10% (w/v) PEG 4000 (apo1 condition), or in 0.1 M ammonium sulfate, 0.05 M magnesium sulfate heptahydrate, 0.1 M sodium citrate pH 5.5% and 22.5 % v/v PEG Smear Medium (BCS screen from Molecular Dimensions) (apo2 condition). These crystals led to the apo structures (apo1 and 2, respectively). No sign of the sulfopeptide was observed in the electron density. Consequently, new crystals were produced in the apo1 crystallization condition in order to perform crystal soaking experiments. A single crystal was obtained and broken into three pieces, each of which were incubated in mother liquor supplemented with either p-cresol sulfate, CCR2 or ACKR1 sulfopeptide (all in powder form) for 3 days. Crystals were then frozen in liquid nitrogen after being soaked in mineral oil. Diffraction data were recorded on the MASSIF-1 beamline at the ESRF, Grenoble, France (soaks), or on the X06DA beamline at the Swiss Light Source (apo 1 and 2). All data were automatically processed by xia2 (*Winter et al., 2013*), and scaled intensities from XDS (*Kabsch, 2010*) were merged using AIMLESS (*Evans and Murshudov, 2013*) using appropriate high resolution cut-offs to yield the final datasets.

Structural determination was initiated by molecular replacement using the previously published LukE structure (PDB ID: 3ROH) as a search model in PHASER (*McCoy et al., 2007*). The solution was subjected to repetitive rounds of restrained refinement in PHENIX (*Adams et al., 2010*) and Autobuster (*Blanc et al., 2004*) and manual building in COOT (*Emsley et al., 2010*). TLS parameters were included in the final round of refinement. The CCP4 program suite (*Winn et al., 2011*) was used for coordinate manipulations. The structures were validated with Molprobity (*Chen et al., 2010*). Refinement statistics are given in *Table 1*, and final refined coordinates and structure factors have been

deposited in the PDB with accession codes 7P8T (apo 1), 7P8S (apo 2), 7P8U (p-cresol sulfate soak), 7P8X (CCR2 [25]DsYDsYG[29] soak), and 7P93 (ACKR1 [34]DSFPDGDsYGANLE[46] soak).

## Protein–protein docking

Computational docking of LukE onto ACKR1 and CCR5 was performed using the information-driven docking software HADDOCK 2.4, which is available as a webserver (*van Zundert et al., 2016*). In the absence of a high-resolution structure for ACKR1, we used an ensemble of six MD simulation-derived models that have been partially validated using H/DX mass spectrometry data in our previous ACKR1 study (ref biorxiv). Each model was docked to the LukE apo1 structure using default parameters (six runs). We used as active residues Cys51, Glu202, and Arg267 of ACKR1, based on the work by Spaan and colleagues (*Spaan et al., 2015*). The active residues of LukE were defined as Tyr96, Asp197, Pro215, and Arg275, which are located respectively in loop 1, 2, 3, and 4. Each of these loops contain residues that are divergent between different leukotoxins and that have been implicated in LukED hemolysis in *Vasquez et al., 2020* and/ or in *Peng et al., 2018*. In the case of CCR5 for which several experimental structures are available, the CCR5 chain was extracted from 5UIW (CCR5-CCL5 complex) or 6MEO (CCR5- gp120 – CD4 complex). Each CCR5 chain was docked to the LukE apo1 or apo2 structure with HADDOCK (4 runs). The CCR5 active residues were Lys171, Glu172, Ser179, Arg274 and Asp276, which were found to mediate LukED toxicity in CCR5[+] cells by *Tam et al., 2016*. Tyr96 and Arg275 of LukE were selected as active residues based on *Reyes-Robles et al., 2013*; *Tam et al., 2016*.

For both receptors, the N-terminal region (residues 1–50 of ACKR1 and residues 1–20 of CCR5) were omitted from the models in order to avoid potential steric clashes with the toxin. The best scoring model from the top HADDOCK cluster of the best scoring run was then selected for molecular dynamics refinement.

## Classical molecular dynamics simulations

Prior to MD simulations, the N-terminal region of the receptors (residues 8–50 of ACKR1 and residues 1–20 of CCR5) were added to the selected models, with ACKR1 Tyr41, and CCR5 Tyr3, 10, 14, and 15 in the sulfated form. The ACKR1-LukE and CCR5-LukE MD systems were then set up using the CHARMM-GUI membrane builder (*Wu et al., 2014*). The starting models were each inserted into a hydrated, equilibrated bilayer composed of approximately 500 2-Oleoyl-1-palmitoyl-sn-glycero-3 -phosphocholine (POPC) and 100 cholesterol molecules. Sodium and chloride ions were added to neutralize the system, reaching a final concentration of approximately 150 mM. Molecular dynamics calculations were performed in GROMACS 2020 using the CHARMM36m force field (*Huang et al., 2017*) and the CHARMM TIP3P water model. Forcefield parameters for sulfotyrosine were obtained by combining CHARMM topologies from related molecules. The input systems were subjected to energy minimization, equilibration and production simulation using the GROMACS input scripts generated by CHARMM-GUI (*Lee et al., 2016*). The temperature and pressure were held at 310.15 K and 1 bar, respectively. During production simulations an NPT ensemble was used with semi-isotropic pressure coupling via the Parrinello-Rahman barostat method while the Nose-Hoover thermostat was used to maintain a temperature of 310.15 K. A leapfrog integration scheme was used, and all bonds were constrained allowing for a time-step of 2 ps to be used during NPT equilibration and production MD simulations. For each system, we performed three production runs and subsequently analyzed the resulting trajectories using GROMACS tools to yield iRMSD, complex BSA, number of interchain hydrogen bonds and inter-residue distances.

## AWH molecular dynamics simulations

The AWH method (*Lindahl et al., 2014*) is an adaptive biasing method implemented in GROMACS (*Hess et al., 2008*). This method enables the calculation of PMF profiles along (a) chosen reaction coordinate(s), and functions by applying a time-dependent biasing potential that flattens free energy barriers along the reaction path. In the case of ACKR1-LukE, the sTyr 41 – Arg263 and sTyr41 – Tyr269 COM distances were defined as a two-dimensional reaction coordinate to report on site 1 and site 2 sulfotyrosine interactions. The sampling interval was 0.2–3.5 nm. The force constant and initial error for AWH calculations were set to 10,000 kJ/mol/nm$^2$ and 10 kJ/mol, respectively. An estimated diffusion parameter of $1.10^{-5}$ nm$^2$/ps was used for each coordinate dimension. A free energy cutoff of

50 kJ/mol was applied to the AWH target distribution to avoid sampling of very high free energy regions. A single trajectory of 2.147 µs was calculated in GROMACS. The free energy profiles at different simulation times were constructed using the gmx awh program included in GROMACS. The final PMF seemed converged with no major change taking place after about 1.8 µs when it exited the initial stage. The estimated AWH target error at the end of the simulation was ~2.2 kJ/mol (down from 10 kJ/mol at the start). In the case of CCR5-LukE, the sTyr 10 – Arg263 and sTyr10 – Tyr269 COM distances were defined as a two-dimensional reaction coordinate. Simulation parameters were the same as for the ACKR1-LukE system. A single trajectory of 2.050 µs was calculated in GROMACS. The simulation exited the initial stage and started converging quickly after sampling its global minimum multiple times in the first 100 ns.

## Acknowledgements

This work was supported by the french Agence Nationale de la Recherche (project ANR-17-CE15-0002-01, CHEMSPEC). The authors would like to thank the staff of beamline BM29 and MASSIF-1 at the European Synchrotron Radiation Facility (Grenoble, France) for assistance with X-ray data collection. The authors would also like to thank the staff of beamline X06DA at the PSI for assistance with crystal testing and data collection. The FP7 WeNMR (project# 261572), H2020 West-Life (project# 675858), the EOSC-hub (project# 777536) and the EGI-ACE (project# 101017567) European e-Infrastructure projects are acknowledged for the use of their web portals, which make use of the EGI infrastructure with the dedicated support of CESNET-MCC, INFN-PADOVA-STACK, INFN-LNL-2, NCG-INGRID-PT, TW-NCHC, CESGA, IFCA-LCG2, UA-BITP, SURFsara and NIKHEF, and the additional support of the national GRID Initiatives of Belgium, France, Italy, Germany, the Netherlands, Poland, Portugal, Spain, UK, Taiwan and the US Open Science Grid. Peptide synthesis was performed using Synbio3 platform supported by GIS IBISA and ITMO Cancer (Montpellier, France). We would like to thank Pr Berk Hess for help in setting up the initial AWH simulations. We additionally would like to thank Marc Leyrat for his help in making Figure 6 - Video 1 and Figure 7 - Video 1.

## Additional information

### Funding

| Funder | Grant reference number | Author |
| --- | --- | --- |
| Agence Nationale de la Recherche | ANR-17-CE15-0002-01 | Cédric Leyrat |

The funders had no role in study design, data collection and interpretation, or the decision to submit the work for publication.

### Author contributions

Paul Lambey, Investigation, Validation; Omolade Otun, Formal analysis, Investigation, Validation, Visualization, Writing – review and editing; Xiaojing Cong, Formal analysis, Investigation, Resources; François Hoh, Investigation, Project administration, Validation; Luc Brunel, Pascal Verdié, Resources; Claire M Grison, Fanny Peysson, Sylvain Jeannot, Investigation; Thierry Durroux, Formal analysis, Supervision, Validation; Cherine Bechara, Formal analysis, Investigation, Writing – review and editing; Sébastien Granier, Conceptualization, Investigation, Project administration, Supervision, Validation, Writing – review and editing; Cédric Leyrat, Conceptualization, Formal analysis, Funding acquisition, Investigation, Methodology, Project administration, Supervision, Validation, Visualization, Writing - original draft, Writing – review and editing

### Author ORCIDs

Pascal Verdié ⓘ http://orcid.org/0000-0002-5807-0293
Sébastien Granier ⓘ http://orcid.org/0000-0003-1550-3658
Cédric Leyrat ⓘ http://orcid.org/0000-0003-0189-0562

### Decision letter and Author response

Decision letter https://doi.org/10.7554/eLife.72555.sa1

Author response https://doi.org/10.7554/eLife.72555.sa2

## Additional files

### Supplementary files

• Transparent reporting form

### Data availability

Diffraction data have been deposited in PDB under the accession codes 7P8T, 7P8S, 7P8U, 7P8X and 7P93. Source Data files containing the computational models of the ACKR1-LukE and CCR5-LukE complexes in Figures 6 and 7 have been provided as pdb files. Figure 2—source data 1 contain the numerical data used to generate the figure.

The following datasets were generated:

| Author(s) | Year | Dataset title | Dataset URL | Database and Identifier |
|---|---|---|---|---|
| Lambey P, Hoh F, Granier S, Leyrat C | 2022 | Crystal Structure of leukotoxin LukE from *Staphylococcus aureus* at 1.5 Angstrom resolution | https://www.rcsb.org/structure/7P8T | RCSB Protein Data Bank, 7P8T |
| Lambey P, Hoh F, Granier S, Leyrat C | 2022 | Crystal Structure of leukotoxin LukE from *Staphylococcus aureus* at 1.9 Angstrom resolution | https://www.rcsb.org/structure/7P8S | RCSB Protein Data Bank, 7P8S |
| Lambey P, Hoh F, Peysson F, Granier S, Leyrat C | 2022 | Crystal Structure of leukotoxin LukE from *Staphylococcus aureus* in complex with p-cresyl sulfate | https://www.rcsb.org/structure/7P8U | RCSB Protein Data Bank, 7P8U |
| Lambey P, Hoh F, Peysson F, Granier S, Leyrat C | 2022 | Crystal Structure of leukotoxin LukE from *Staphylococcus aureus* in complex with a doubly sulfated CCR2 N-terminal peptide | https://www.rcsb.org/structure/7P8X | RCSB Protein Data Bank, 7P8X |
| Lambey P, Hoh F, Peysson F, Granier S, Leyrat C | 2022 | Crystal Structure of leukotoxin LukE from *Staphylococcus aureus* in complex with a sulfated ACKR1 N-terminal peptide | https://www.rcsb.org/structure/7P93 | RCSB Protein Data Bank, 7P93 |

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
