## [Editor Report]

In this report, the authors employed a wide array of biophysical tools and techniques to study the interaction of LukE toxin with chemokine receptors. Although follow-up studies will be required to substantiate all of the conclusions drawn by the authors, the paper will be of general interest as it provides insights into the molecular recognition mechanism of an important toxin interacting with cellular receptors.

---

## [Decision Letter]

**Decision letter after peer review:**

Thank you for submitting your article "Structural insights into recognition of chemokine receptors by *Staphylococcus aureus* leukotoxins" for consideration by *eLife*. Your article has been reviewed by 3 peer reviewers, one of whom is a member of our Board of Reviewing Editors, and the evaluation has been overseen by Olga Boudker as the Senior Editor. The reviewers have opted to remain anonymous.

Essential revisions:

The reviewers agree that while the study is quite extensive in its scope and clearly deals with an important biomedical phenomenon, there are multiple concerns about the study which the authors would have to clarify through additional experiments. For example, the reviewers agree that the manuscript needs further experimental studies using biophysical techniques to validate computational models. As concluded by the authors, models based on a combination of crystallographic and mutagenesis data may be further improved by biophysical data, as for example derived from HDX-MS or NMR. Including such experimental data would strengthen the manuscript.

Specific Comments to the Authors:

1. Regarding the MD simulations:

a. What was the rationale for simulation using two force fields? Was it simply to generate diverse conformational libraries for EROS?

b. The authors carried out AWH MD simulations of the ACKR1-LukE, what about the CCR5-LukE system? The authors should include these results or explain why it is missing.

c. Related to this, a more concise presentation of the MD simulations would be to only present AWH MD simulations of the ACKR1-LukE and CCR5-LukE. The conventional MD results don't add much to the story. The author might even consider removing the results and figures associated with conventional MD or putting them in supporting information less dramatically. Using the AWH simulations and their PMFs, the authors can harvest and show the snapshots at the dominant local minima and use those structures as the MD-based models of these complexes. And instead of analyzing the structures in the final 300 ns (Figure 5), which is a bit arbitrary, analyze and show the details of the most probable structures.

2. In the abstract, would it be possible to include the common names for erythrocytes and leukocytes in parentheses, i.e., erythrocytes (red blood cells) and leukocytes (white blood cells)? This change might make the abstract even more accessible to a broader audience.

9. X-ray structures of LukE: Of the multiple high-resolution structures shown by the authors the apo and p-cresol sulfate bound structures are fairly convincing. The structures in complex the two receptor peptides "DSFPDGDsYGANLE" and "DsYDsYG" have multiple issues.

a. The longer peptide complex shows very weak 2FoFc density and I can see density clearly only when I contour the map at a level of 0.4 which is very low. At the usual contour level that one uses for a structure of this resolution, say around 0.8 to 1, there are serious breaks in the density and only the sulfated tyrosine sidechain looks reasonable.

b. The authors make no attempt to show the omit density for any of the bound ligands (a standard practice) and do not comment on the weak quality of the peptide density.

c. The DSFPDGDsYGANLE peptide is also given an occupancy of just 0.3 which is very low and I am therefore not very convinced about this peptide binding site.

d. Moreover, there are two of these peptides bound at site 1 and site 2 and the authors do not make it clear to which of the two peptides binding sites they think is the binding site for the native receptor, before proceeding to explain the structure with double sulfated tyrosine peptide complex.

e. The density for the peptide "DsYDsYG" from CCR2 is only marginally better (again modeled at low occupancy) and here also there are breaks in 2FoFc map at the normal contour levels. The peptide however interacts with both site 1 and 2 suggesting that this could be a stronger interaction. The lack of omit or polder maps in the manuscript for the ligand-bound LukE is quite surprising.

10. LukE peptide interaction affinities: Although the authors have performed crystallographic analysis, there would be much more confidence if binding assays of the two peptides and p-cresol sulfate are done with LukE to obtain affinity and stoichiometry of the interaction. The ability of the CCR2 double sulfated peptide interacting at sites 1 and 2 could further serve as a basis to understand the high affinity of LukE to CCR2 in comparison to AckR1.

11. Docking and interaction analysis: While the authors have generated their docking based on prior mutant information and the docked model seems largely convincing, I hope to see some low-resolution experimental data to be convinced about this organization. Can the authors potentially look to obtain a low-resolution electron microscopy image or 2D classes (a fairly routine exercise) of LukE complexed with CCR2 to further validate their modeled complex? This will go a long way in enhancing the confidence in the findings.

12. Also, can the authors clarify the effects of mutating Arg275 of LukE on its ability to interact with the receptor? If the interaction interface is accurate, mutating this residue should significantly weaken this interaction.

13. Authors referenced previous SPR studies characterizing the interaction between LukE and CCR5/ACKR1 receptors and could also use a similar approach to validate computational models using a carefully selected set of point mutations.

14. TR-FRET assay could be used to evaluate the effect of mutations in LukE disrupting the complexes with cytokine receptors.

15. Crystal structures of the complex between LukE and ligands may be biased by crystal pacing and may also reflect non-native interactions when soaked at high ligand concentrations. Binding sites for sulfated tyrosines should be validated experimentally, for example using NMR.

---

## [Author Response]

Essential revisions:The reviewers agree that while the study is quite extensive in its scope and clearly deals with an important biomedical phenomenon, there are multiple concerns about the study which the authors would have to clarify through additional experiments. For example, the reviewers agree that the manuscript needs further experimental studies using biophysical techniques to validate computational models. As concluded by the authors, models based on a combination of crystallographic and mutagenesis data may be further improved by biophysical data, as for example derived from HDX-MS or NMR. Including such experimental data would strengthen the manuscript.Specific Comments to the Authors:1. Regarding the MD simulations:a. What was the rationale for simulation using two force fields? Was it simply to generate diverse conformational libraries for EROS?

The rationale for using 2 force fields in the MD simulations of LukE was indeed to generate more diverse conformations of the protein. The AMBER99SB-ILDN* force field tends to favor collapsed states for the intrinsically disordered regions (signal peptide and polyhistidine tag in this case), while the AMBER99SBWS force field samples more effectively the extended conformations. We have clarified this point for the reader by adding this information in the methods section.

b. The authors carried out AWH MD simulations of the ACKR1-LukE, what about the CCR5-LukE system? The authors should include these results or explain why it is missing.

The reason why we didn’t perform the AWH MD simulations of the CCR5-LukE system is because we thought that the classical MD simulations were already pretty convincing for that system (given that all 3 MD trajectories showed binding of sTyr10 at site 1 on the toxin). In the case of ACKR1-LukE, only two out of three MD trajectories showed binding of sTyr41 at site 1, so we deemed it worthwhile to use AWH MD to further validate the model.

Nevertheless, we have now produced a 2.05 µs AWH MD trajectory of CCR5-LukE, which confirms the results obtained in classical MD, showing binding of sTyr10 at site 1 and sTyr14 at site 2 at the dominant local energy minimum. The results are shown in figure 8, and the results and methods sections have been updated accordingly.

c. Related to this, a more concise presentation of the MD simulations would be to only present AWH MD simulations of the ACKR1-LukE and CCR5-LukE. The conventional MD results don't add much to the story. The author might even consider removing the results and figures associated with conventional MD or putting them in supporting information less dramatically. Using the AWH simulations and their PMFs, the authors can harvest and show the snapshots at the dominant local minima and use those structures as the MD-based models of these complexes.

Although the idea of using the snapshots from the dominant local minimum of the AWH simulations is attractive, we think the conventional MD model shows better agreement with the experimental data at the level of the orthosteric pocket interactions with LukE. This is true at least for the ACKR1LukE system, where we observed that the conventional MD model forms a tightly packed hydrophobic cluster involving Tyr279, Phe273 from LukE and Cys51-Cys276 from ACKR1 (which are SS linked). This is consistent with the study by (Spaan et al. ,2015) who found that Cys51 to Ser mutation in ACKR1 had a strong impact on LukED activity (but not HlgAB). On the contrary, in the AWH-extracted models of ACKR1-LukE, the disulfide bridged Cys51-Cys276 are not interacting with LukE because of conformational rearrangements at the receptor-toxin interface. Similarly, when we compare conventional MD and AWH models of CCR5-LukE, the toxin seems to insert less deeply into the orthosteric pocket in the AWH models.

Our AWH simulations are designed to calculate the potential of mean force along the chosen reaction coordinate, which corresponds to distances between ACKR1 sTyr41 (or CCR5 sTyr10) and LukE Arg263 or Tyr269 located at site 1 and 2. The bias introduced to the simulation greatly increase the conformational sampling along the reaction coordinate and effectively reconstitutes the PMF, however it is possible that this extra energy may indirectly destabilize other regions of the protein complex (such as the orthosteric pocket interaction site). It is also possible that the most stable conformation that can be reached for the sulfotyrosine site 1 interaction would be incompatible with the most stable conformation of the orthosteric pocket interface, putting the complex in a dynamic equilibrium. In any case, there is no theoretical reason for the most stable AWH models to represent the most stable state of the orthosteric pocket interaction site because the chosen reaction coordinate doesn’t report on this region of the receptor-toxin complex. For these reasons, we believe that conventional and AWH MD approaches are complementary and we do not want to emphasize one over the other.

And instead of analyzing the structures in the final 300 ns (Figure 5), which is a bit arbitrary, analyze and show the details of the most probable structures.

We agree with the reviewer that analyzing the most probable structures would indeed be a more rigorous approach. We have performed clustering analysis on the conventional MD trajectories of ACKR1-LukE and CCR5-LukE, and selected the middle structure of the largest cluster as MD-based models of these complexes (RMSD of about 1Å with the models previously used). The close ups in figure 5 and 7 have been updated accordingly. This sentence was added to the figure legends “The MD model of ACKR1-LukE (or CCR5-LukE in Figure 7) corresponds to the middle structure of the largest cluster in the MD trajectory ”. The models resulting from the clustering analysis have been added as source data to Figure 5 and 7.

8. In Figure 1B, where did the full signal peptide come from? It does not appear in the crystal structure, was it modeled?

The signal peptide (without residues 1 to 11, not present in the construct) was modeled in order to generate a conformational ensemble of LukE and fit the SAXS data. We started from the APO2 structure in which residues 12-23 were already visible, and added the missing residues (residues 2428, but also the C-terminal polyhistidine tag). When fitting SAXS data using ensembles of models, it is critical that the model has all the residues that were present in the construct including tags and disordered regions. This information has been added to the methods section (Molecular dynamics and ensemble optimization of LukE).

9. X-ray structures of LukE: Of the multiple high-resolution structures shown by the authors the apo and p-cresol sulfate bound structures are fairly convincing. The structures in complex the two receptor peptides "DSFPDGDsYGANLE" and "DsYDsYG" have multiple issues.a. The longer peptide complex shows very weak 2FoFc density and I can see density clearly only when I contour the map at a level of 0.4 which is very low. At the usual contour level that one uses for a structure of this resolution, say around 0.8 to 1, there are serious breaks in the density and only the sulfated tyrosine sidechain looks reasonable.

We think part of the issue is because we performed all the structure refinement in Buster but we ran a quick refinement in Phenix on the final "DSFPDGDsYGANLE" and "DsYDsYG" structures. After initial submission, we were asked to provide pdb and mtz files for each structure, and we uploaded both the Phenix and Buster refinements. We have produced Author response image 1 with Coot snapshots of the different maps for the "DSFPDGDsYGANLE" structure. In Buster, the "DSFPDGDsYGANLE" structure has reasonable density for the 2 peptide chains (B and M), although both chains have occupancies of only 0.3 to 0.6 (with the strongest signal for the sulfotyrosines at 0.66 and 0.69 occupancy). There is only a small chain break when the map is contoured at 1σ at Gly42 of chain M. however, the final Buster structure still has a lot of positive signal in the Fo-Fc difference map at 3σ, but most of the signal vanishes at 4σ (panels B and C in the figure). We think this signal comes from structural heterogeneity in the bound conformation of the peptide, such as the presence of alternative minor populations. We then decided to run a refinement using Phenix, which has a different method for computing the electron density maps compared to Buster – in order to see if the signal in the difference map is also present. The Phenix refined structure shows no significant signal in the difference map at 3σ, but the electron density for peptide chain M is very faint in the 2Fo-Fc map, except for the sulfotyrosine (panel A of Author response image 1).

**Author response image 1. sa2fig1:** 

b. The authors make no attempt to show the omit density for any of the bound ligands (a standard practice) and do not comment on the weak quality of the peptide density.

We have now produced a figure that shows the Polder maps contoured at 2.5 σ for each structure. We have also added some text to comment on the weak quality of the peptide density. “Although the signal for each sulfotyrosine is fairly strong, the electron density of the other peptide residues is weaker, particularly for the 1^st^ peptide (see Polder map in figure S4B). This is apparent in the refined occupancies which are ~ 0.7 for the sulfotyrosines, but only 0.3 to 0.6 for the other peptide residues.”

c. The DSFPDGDsYGANLE peptide is also given an occupancy of just 0.3 which is very low and I am therefore not very convinced about this peptide binding site.

We agree that parts of the peptide have a low occupancy value of 0.3. It was however still possible to build them using the high quality maps from Buster, and we are fairly confident in the quality of our model, also given the high resolution of the dataset. It is our experience with a variety of protein crystals that Buster tends to produce better quality maps than other refinement programs, which sometimes is critical when dealing with weak or ambiguous ligand density. As can be seen in panel A of Author response image 1, it might not have been possible to build anything beyond the sulfotyrosine of chain M using the maps from Phenix.

d. Moreover, there are two of these peptides bound at site 1 and site 2 and the authors do not make it clear to which of the two peptides binding sites they think is the binding site for the native receptor, before proceeding to explain the structure with double sulfated tyrosine peptide complex.

We agree with the reviewer that we didn’t make this point particularly clear. The peptide that is bound at site 2 and has the highest overall occupancy, is running upside down on the toxin surface (the C-terminus is going away from the direction of the membrane surface). In addition, a symmetry-related LukE molecule also interacts with the sulfate group of the sulfotyrosine. The peptide bound at site 1 has is C-terminus going toward the membrane and doesn’t interact with any symmetry related molecule. However, its conformation can still be influenced by crystal packing and obviously by the presence of the other peptide molecule as well. For these reasons, we think that the conformations of the peptides don’t necessarily carry biological meaning (but maybe only biophysical meaning), and that only the location of the sulfotyrosine binding sites can be inferred with confidence from these structures. We have added some text to clarify this point. “Although the conformation of both peptides is stabilized by multiple interactions with LukE, it should be noted that the 2^nd^ peptide chain runs with its C-terminus going away from the putative location of the membrane. In addition, the occupancy is relatively low and the conformation of the peptides is likely influenced by crystal packing and the associated steric constraints.”

e. The density for the peptide "DsYDsYG" from CCR2 is only marginally better (again modeled at low occupancy) and here also there are breaks in 2FoFc map at the normal contour levels. The peptide however interacts with both site 1 and 2 suggesting that this could be a stronger interaction. The lack of omit or polder maps in the manuscript for the ligand-bound LukE is quite surprising.

Similar to the other peptide structure, the maps obtained from Buster for the "DsYDsYG" structure show a stronger density for the peptide compared to Phenix. The occupancy is 0.98 for site 1 sulfotyrosine (2 alternate conformations modelled, 0.49 occupancy each) and 0.56 for site 2 sulfotyrosine. There also seems to be a lot of conformational heterogeneity for this bound peptide, with at least 2 distinct possibilities as we show in figure S4 C and D. There is one chain break at Asp27, and we believe this is due to this residue being disordered because of the conformational equilibrium of the peptide adopting different bound conformations. We have added a figure with the POLDER omit map contoured at 2.5 σ, to complement the 2Fo-Fc map at 1σ shown in figure S4. And we have also added some text to comment on the occupancy and weak electron density. “This is consistent with the lower occupancy for site 2 versus site 1 sulfotyrosine (0.56 and 0.98), and the presence of a chain break at Asp27 in the electron density (Figure S3). These observations indicate that the binding of the doubly sulfated CCR2 peptide is highly dynamic within the crystal.”

10. LukE peptide interaction affinities: Although the authors have performed crystallographic analysis, there would be much more confidence if binding assays of the two peptides and p-cresol sulfate are done with LukE to obtain affinity and stoichiometry of the interaction. The ability of the CCR2 double sulfated peptide interacting at sites 1 and 2 could further serve as a basis to understand the high affinity of LukE to CCR2 in comparison to AckR1.

We agree with the reviewer that it would be interesting to use binding assays in order to determine the affinity and stoichiometry of the interaction between LukE and the peptides, or p-cresol sulfate. Such data could also help understand if the interaction at site 1 and 2 for the CCR2 peptide is responsible for the more effective inhibition of CCL2 binding on CCR2 in the TR-FRET assay. We are currently running HDX-MS combined with native MS analysis to explore the effect of the presence of the peptides on toxins dynamics and map binding sites. Our preliminary data confirms peptide binding, however this binding does not only impact the HDX of the direct binding sites but also leads to important conformational changes within the toxins, which we are currently analysing. In addition, binding of the peptides might influence toxin’s propensity of oligomerise, which makes it more complicated to quantify an affinity to one binding site. We do not wish to include this data in the manuscript because it would add unnecessary complexity.

11. Docking and interaction analysis: While the authors have generated their docking based on prior mutant information and the docked model seems largely convincing, I hope to see some low-resolution experimental data to be convinced about this organization. Can the authors potentially look to obtain a low-resolution electron microscopy image or 2D classes (a fairly routine exercise) of LukE complexed with CCR2 to further validate their modeled complex? This will go a long way in enhancing the confidence in the findings.

The characterization of the chemokine receptors in complex with LukE (and other family members) using electron microscopy has been challenging for several reasons. LukE tends to interact with the carbohydrate-based gel filtration column matrices, and also has some affinity for the detergent molecules that are used to solubilize the receptor (DDM, LMNG…), which makes complex purification difficult using standard approaches. The complexes can however be isolated using affinity tag based purifications, but the yield is generally low due to partial dissociation during the washing steps. We did obtain some cryoEM images of ACKR1-LukE on a TALOS microscope, but small size and particle heterogeneity precluded analysis. We have also analysed a CCR5-LukE sample using negative stain EM, which is shown in Author response image 2. Although we could identify particles with the expected size and shape, the quality of the data is poor and we don’t think it deserves to be included in this manuscript.

12. Also, can the authors clarify the effects of mutating Arg275 of LukE on its ability to interact with the receptor? If the interaction interface is accurate, mutating this residue should significantly weaken this interaction.13. Authors referenced previous SPR studies characterizing the interaction between LukE and CCR5/ACKR1 receptors and could also use a similar approach to validate computational models using a carefully selected set of point mutations.14. TR-FRET assay could be used to evaluate the effect of mutations in LukE disrupting the complexes with cytokine receptors.

We have designed 4 mutants of LukE to disrupt the interactions with CCR5/ACKR1 at the orthosteric pocket and sulfotyrosine binding site 1 and 2, which we evaluated using the TR-FRET assay:

The R85G-R290E mutant neutralizes sulfotyrosine site 1 and favors intramolecular side chain interactions between R263 and mutated E290.The Y269E mutant disrupt sulfotyrosine site 2. The mutation favors intramolecular side chain interactions between R101/K283 and mutated E269, destabilizing the intermolecular interactions with CCR5 sTyr14 or ACKR1 Asn44/Glu46.The Y279D mutant disrupts the interaction with Glu262/Asn267 in the CCR5-LukE model, and with Cys51-Cys276 in the ACKR1-LukE model.The F273D-R275D disrupts multiple interactions at the orthosteric pocket site, including

R275-D276/E283 in CCR5-LukE and R275-D263/E290 in ACKR1-LukE. In addition, in ACKR1LukE, F273 is part of an hydrophobic cluster that includes Y279/F273 of LukE and Cys51Cys276 of ACKR1.

The activity of the mutants in the TR-FRET binding assay was abolished or strongly decreased, providing an additional validation of our models. The results are shown in figure 9. Supplementary figures 8 and 9 show the environment of each mutated residues in the CCR5 and ACKR1 models.

15. Crystal structures of the complex between LukE and ligands may be biased by crystal pacing and may also reflect non-native interactions when soaked at high ligand concentrations. Binding sites for sulfated tyrosines should be validated experimentally, for example using NMR.

We acknowledge that the crystal structures of LukE in complex with ligands represent an artificial system which is constrained by crystal packing interactions, and that non-native interactions can be induced by high concentrations of soaked ligands. However, we believe the sulfotyrosine binding sites have been adequately validated through multiple techniques:

3 crystal structures with ligands of different sizes (and supposedly a differential impact of the crystal packing for each ligand).Spontaneous binding of the sulfotyrosines in MD simulations for 2 different systems (and in multiple independent MD trajectories), and validation of site 1 using AWH simulations.The new TR-FRET assays with sulfotyrosine site 1 and site 2 mutants showing decreased or abolished binding.Mutations of ACKR1 tyrosine 41 already described in the literature (Grison et al., 2021, Spaan et al., 2015).